# Cannabinoid signaling modulation through JZL184 restores key phenotypes of a mouse model for Williams–Beuren syndrome

Alba Navarro-Romero[1†], Lorena Galera-López[1†], Paula Ortiz-Romero[2], Alberto Llorente-Ovejero[3], Lucía de los Reyes-Ramírez[1], Iker Bengoetxea de Tena[3], Anna Garcia-Elias[1], Aleksandra Mas-Stachurska[4], Marina Reixachs-Solé[5,6], Antoni Pastor[7], Rafael de la Torre[7], Rafael Maldonado[1,7], Begoña Benito[8,9,10], Eduardo Eyras[5,6,7], Rafael Rodríguez-Puertas[3,11], Victoria Campuzano[2], Andres Ozaita[1]*

[1]Laboratory of Neuropharmacology, Department of Medicine and Life Sciences, Universitat Pompeu Fabra, Barcelona, Spain; [2]Department of Biomedical Sciences, School of Medicine and Health Sciences, University of Barcelona, and centro de Investigación Biomédica en Red de Enfermedades Raras (CIBERER), Barcelona, Spain; [3]Department of Pharmacology, Faculty of Medicine and Nursing, University of the Basque Country, Leioa, Spain; [4]Hospital del Mar Medical Research Institute (IMIM), Autonomous University of Barcelona, Barcelona, Spain; [5]EMBL Australia Partner Laboratory Network at the Australian National University, Canberra, Australia; [6]The John Curtin School of Medical Research, Australian National University, Canberra, Australia; [7]Hospital del Mar Medical Research Institute (IMIM), Barcelona, Spain; [8]Group of Cardiovascular Experimental and Translational Research (GET-CV), Vascular Biology and Metabolism, Vall d'Hebron Research Institute (VHIR),, Barcelona, Spain; [9]Department of Medicine, Universitat Autònoma de Barcelona, Barcelona, Spain; [10]Centro de Investigación Biomédica en Red de Enfermedades Cardiovasculares (CIBER-CV), Instituto de Salud Carlos III, Madrid, Spain; [11]Neurodegenerative Diseases, Biocruces Bizkaia Health Research Institute, Barakaldo, Spain

*For correspondence:
andres.ozaita@upf.edu

†These authors contributed equally to this work

**Abstract** Williams–Beuren syndrome (WBS) is a rare genetic multisystemic disorder characterized by mild-to-moderate intellectual disability and hypersocial phenotype, while the most life-threatening features are cardiovascular abnormalities. Nowadays, there are no pharmacological treatments to directly ameliorate the main traits of WBS. The endocannabinoid system (ECS), given its relevance for both cognitive and cardiovascular function, could be a potential druggable target in this syndrome. We analyzed the components of the ECS in the complete deletion (CD) mouse model of WBS and assessed the impact of its pharmacological modulation in key phenotypes relevant for WBS. CD mice showed the characteristic hypersociable phenotype with no preference for social novelty and poor short-term object-recognition performance. Brain cannabinoid type-1 receptor (CB1R) in CD male mice showed alterations in density and coupling with no detectable change in main endocannabinoids. Endocannabinoid signaling modulation with subchronic (10 days) JZL184, a selective inhibitor of monoacylglycerol lipase, specifically normalized the social and cognitive phenotype of CD mice. Notably, JZL184 treatment improved cardiovascular function and restored gene expression patterns in cardiac tissue. These results reveal the modulation of the ECS as a promising novel therapeutic approach to improve key phenotypic alterations in WBS.

## Editor's evaluation

In this report, alterations of the brain cannabinoid type-1 receptor (CB1R) were identified in a mouse model of Williams-Beuren syndrome (CD mice). Modulation of CB1R by JZL184 treatment improved social and cognitive phenotypes and also cardiac function of CD mice. These are important and novel findings highlight a role of the endocannabinoid system and its dysfunction in Williams-Beuren syndrome (WBS) that provide a basis for future developments of potential therapeutics for WBS patients. This study will be of great interest to researchers and clinicians in the field of human genetic diseases.

## Introduction

Williams–Beuren syndrome (WBS) is a genetic neurodevelopmental disorder caused by a hemizygous deletion of a region containing 26–28 genes at chromosomal band 7q11.23. The estimated prevalence of this disorder is 1 in 7500 individuals (*Strømme et al., 2002*). Subjects present manifestations affecting mainly the central nervous system and the cardiovascular system (*Pérez Jurado, 2003*; *Kozel et al., 2021*; *Royston et al., 2019*). WBS subjects show mild-to-moderate intellectual disability with an intelligence quotient (IQ) score from 40 to 90 (*Bellugi et al., 2000*; *Mervis and Klein-Tasman, 2000*) affecting their quality of life, where independent living is infrequent (*Howlin and Udwin, 2006*). One of the most prominent features of the cognitive profile of WBS is a hypersociable phenotype characterized by uninhibited social interactions and a reduced response to social threat (*Gosch and Pankau, 1994*; *Plesa-Skwerer et al., 2006*). This phenotype is opposite to the archetypic social phenotype of autism spectrum disorders (ASDs) characterized by lack of social interest and deficits in social communication (*Barak and Feng, 2016*). Notably, the congenital cardiovascular phenotype in WBS is the major source of morbidity and mortality (*Collins, 2018*), characterized by elastin arteriopathy, supravalvular aortic stenosis, peripheral pulmonary stenosis, and hypertension, which require in many cases surgical correction (*Pober et al., 2008*). While strategies such as behavioral intervention can improve WBS cognitive skills to some extent, or certainly invasive surgical procedures are available, their success is limited and WBS is largely without treatment (*Morris et al., 2020*).

Several mouse models have been developed mimicking the genetic alterations observed in WBS subjects (*Osborne, 2010*). Among them, the complete deletion (CD) mouse model resembles the most common hemizygous deletion found in WBS patients and displays several WBS phenotypic traits (*Segura-Puimedon et al., 2014*). Indeed, this model shows significant alterations in social behavior with enhanced sociability (*Segura-Puimedon et al., 2014*), cognitive deficits (*Ortiz-Romero et al., 2018*), and a mild cardiovascular phenotype with cardiac hypertrophy, borderline hypertension, and mildly increased arterial wall thickness (*Segura-Puimedon et al., 2014*) among others.

The endocannabinoid system (ECS) is a homeostatic modulatory system involved in a plethora of physiological functions at both central and peripheral levels. It is composed by cannabinoid receptors, including cannabinoid type-1 and cannabinoid type-2 receptors (CB1R and CB2R, respectively), their endogenous ligands or endocannabinoids (mainly, 2-arachidonoylglycerol, 2-AG, and *N*-arachidonoylethanolamine, AEA), and the enzymes involved in the synthesis and inactivation of these ligands. The main enzymes involved in the biosynthesis of 2-AG and AEA are 1,2-diacylglycerol (DAG) and *N*-acyl-phosphatidylethanolamine-specific phospholipase D (NAPE-PLD), respectively, while their degradation is controlled mainly by monoacylglycerol lipase (MAGL), in the case of 2-AG, and fatty acid amide hydrolase (FAAH) in the case of AEA. CB1R and CB2R are both G-protein-coupled receptors (GPCRs) mainly signaling through inhibitory $G_{i/o}$ proteins (*Lutz, 2020*). CB1R is highly expressed in different brain regions, including the hippocampus, the basolateral amygdala, and the prefrontal cortex, where it is mainly located at presynaptic terminals. Endocannabinoids act as retrograde messengers binding to presynaptic CB1R and controlling neurotransmitter release at both excitatory and inhibitory synapses (*Katona and Freund, 2012*). The ECS regulates different behavioral responses including sociability (*Wei et al., 2017*), cognition (*Katona and Freund, 2012*), or emotional responses (*Lutz et al., 2015*), which are usually impaired in neurodevelopmental disorders. In fact, multiple lines of evidence point to the dysregulation of the ECS in the pathophysiology of neurodevelopmental conditions (*Navarro-Romero et al., 2019*; *Busquets-Garcia et al., 2013*; *Carreira et al., 2022*). Interestingly, alterations of the ECS have been described in several mouse models for ASD with altered sociability (*Zamber-letti et al., 2017*). In addition, the pharmacological modulation of the ECS, whether targeting the

**eLife digest** Williams-Beuren syndrome (WBS) is a rare disorder that causes hyper-social behavior, intellectual disability, memory problems, and life-threatening overgrowth of the heart. Behavioral therapies can help improve the cognitive and social aspects of the syndrome and surgery is sometimes used to treat the effects on the heart, although often with limited success. However, there are currently no medications available to treat WBS.

The endocannabinoid system – which consists of cannabis-like chemical messengers that bind to specific cannabinoid receptor proteins – has been shown to influence cognitive and social behaviors, as well as certain functions of the heart. This has led scientists to suspect that the endocannabinoid system may play a role in WBS, and drugs modifying this network of chemical messengers could help treat the rare condition.

To investigate, Navarro-Romero, Galera-López et al. studied mice which had the same genetic deletion found in patients with WBS. Similar to humans, the male mice displayed hyper-social behaviors, had memory deficits and enlarged hearts. Navarro-Romero, Galera-López et al. found that these mutant mice also had differences in the function of the receptor protein cannabinoid type-1 (CB1).

The genetically modified mice were then treated with an experimental drug called JZL184 that blocks the breakdown of endocannabinoids which bind to the CB1 receptor. This normalized the number and function of receptors in the brains of the WBS mice, and reduced their social and memory symptoms. The treatment also restored the animals' heart cells to a more normal size, improved the function of their heart tissue, and led to lower blood pressure. Further experiments revealed that the drug caused the mutant mice to activate many genes in their heart muscle cells to the same level as normal, healthy mice.

These findings suggest that JZL184 or other drugs targeting the endocannabinoid system may help ease the symptoms associated with WBS. More studies are needed to test the drug's effectiveness in humans with this syndrome. Furthermore, the dramatic effect JZL184 has on the heart suggests that it might also help treat high blood pressure or conditions that cause the overgrowth of heart cells.

cannabinoid receptors, or the enzymes involved in the degradation of the endocannabinoids, restores social abnormalities in some of these models (*Wei et al., 2017*; *Zamberletti et al., 2017*). In addition, approaches targeting the ECS have been demonstrated to improve cognitive impairment and plasticity in mouse models of Down syndrome and fragile X syndrome (*Navarro-Romero et al., 2019*; *Busquets-Garcia et al., 2013*). Therefore, nowadays, there is an interest in developing clinical trials to assess the real therapeutic potential of the ECS in neurodevelopmental disorders (*Müller-Vahl et al., 2021*; *Müller-Vahl et al., 2022*).

So far, the role and therapeutic potential of the ECS in social behavior, cognition, and other key phenotypes of WBS had not been addressed before. In this study, we investigated the brain components of the ECS in the WBS CD model to find significant alterations in CB1R expression and G-protein coupling in specific brain regions. Additionally, we reveal that subchronic administration of the MAGL inhibitor JZL184 normalized relevant behavioral phenotypes in CD mice including social behavior and memory alterations. Interestingly, this treatment also partially restored cardiovascular deficits and cardiac transcriptional alterations found in the model. Altogether, our results indicate that the modification of the endocannabinoid signaling could be a novel therapeutic strategy worth evaluating in the context of WBS.

## Results
### CD mice exhibit social and cognitive alterations
We first analyzed social behavior in CD mice and their WT littermates using the Vsocial-maze (*Martínez-Torres et al., 2019*; *Figure 1a*). This test allows to assess exploration, sociability, and preference for social novelty in the same mouse. Exploratory behavior was analyzed in the empty Vsocial-maze during the habituation phase by accounting the time mice explored both empty compartments (E1 and E2) at the end of the corridors. No changes were observed between genotypes (*Figure 1b*, left). Then, during the sociability phase, both WT and CD mice displayed a significant preference for

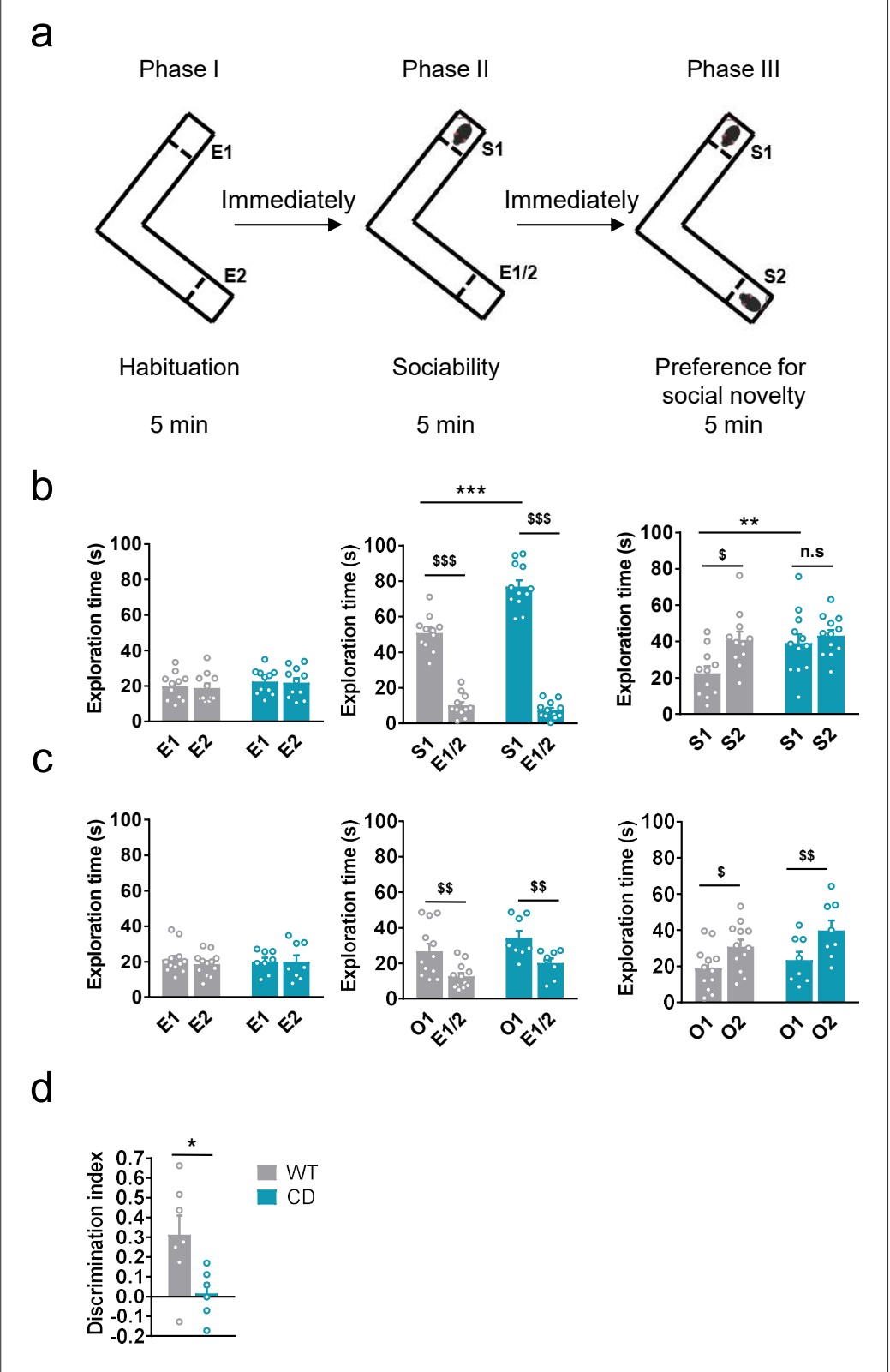

**Figure 1.** Complete deletion (CD) mice show an hypersociable phenotype, no preference for social novelty and cognitive alterations. (**a**) Schematic cartoon of the sociability and preference for social novelty procedure. (**b**) Time spent exploring either empty compartments (E) or stranger mice (S) during the three phases of the Vsocial-maze (WT, *n* = 11; CD, *n* = 11–12). (**c**) Time spent exploring either empty compartments (E) or objects (O) (WT,

*Figure 1 continued on next page*

*Figure 1 continued*

*n* = 12; CD, *n* = 8). Statistical significance was calculated by repeated measures analysis of variance (ANOVA) comparison. $^\$$p < 0.05; $^{\$\$}$p < 0.01; $^{\$\$\$}$p < 0.001 (compartment effect); **p < 0.01; ***p < 0.001 (genotype effect). (**d**) Discrimination index of WT and CD mice (WT, *n* = 7; CD, *n* = 6). Statistical significance was calculated by Student's *t*-test. *p < 0.05 (genotype effect). Data are expressed as mean ± standard error of the mean (SEM).

The online version of this article includes the following source data and figure supplement(s) for figure 1:

**Source data 1.** Complete deletion (CD) mice social and cognitive alterations.

**Figure supplement 1.** Schematic cartoon of behavioral test.

**Figure supplement 2.** Long-term nonemotional, emotional, and spatial memory in complete deletion (CD) mice.

**Figure supplement 2—source data 1.** Long-term nonemotional, emotional, and spatial memory in complete deletion (CD) mice.

exploring a compartment with an unfamiliar juvenile mouse (stranger 1, S1) rather than an empty compartment. Notably, CD mice spent significantly more time than WT mice exploring S1 (*Figure 1b*, middle). Finally, during the preference for social novelty phase, CD mice explored similarly S1 and a novel unfamiliar juvenile mouse (stranger 2, S2) in contrast to WT animals that showed a significant preference for the novel stranger (*Figure 1b*, right). These data indicated that CD mice presented a hypersociable phenotype and a lack of preference for social novelty.

To confirm that both alterations in social behavior were dependent on social stimuli, we repeated the same procedure using unfamiliar objects instead of unfamiliar mice (see setting in *Figure 1— figure supplement 1a*). Both WT and CD mice displayed a preference for the compartment with an object (object 1, O1) over the empty compartment. In contrast to social behavior, WT and CD mice spent similar time exploring O1 (*Figure 1c*, middle). When a new object (object 2, O2) was introduced immediately after to simulate the preference for novelty, both WT and CD mice spent more time exploring the new object O2 than the object that had been previously explored, O1 (*Figure 1c*, right). On the one hand, these results reveal the strong motivation of CD mice for social interaction compared to WT mice. On the other hand, they show both genotypes display similar motivation to explore object novelty. Under our experimental conditions, using the V-maze (see setting in *Figure 1—figure supplement 1b*), CD mice also displayed an impairment in short-term memory in novel object-recognition test (NORT) (*Figure 1d*). Instead, no significant differences were found when the NORT was performed to test long-term memory (*Figure 1—figure supplement 2a*), context fear conditioning (*Figure 1—figure supplement 2b*), or spatial learning and memory in the Barnes maze test (*Figure 1—figure supplement 2c, d*), altogether pointing to specific but not generalized deficiencies in cognitive performance.

## CD mice show alterations in density and signaling of CB1R

We determined the levels of endocannabinoids 2-AG, AEA, and related 2-monoacylglycerols (2-linoleoylglycerol, 2-LG and 2-oleoylglycerol, 2-OG) and *N*-acylethanolamines (*N*-docosatetraenoylethanolamine, DEA and *N*-docosahexaenoylethanolamine, DHEA) in whole brain homogenates. No significant changes were revealed in CD mice in comparison to WT animals (*Table 1*). Then, we analyzed cannabinoid receptor brain density by [³H]CP55,940 radioligand-binding assay in brain tissue sections. Quantitative densitometry revealed an increased density of cannabinoid receptors in the basolateral and central amygdala, whereas a decreased density was observed in the polymorphic and granular layers of dentate gyrus (*Figure 2a, b*; *Supplementary file 1*). To determine the specific

**Table 1.** Levels of endocannabinoids and related compounds in whole brain homogenates of complete deletion (CD) and WT.

(WT, *n* = 10; CD, *n* = 11). Statistical significance was calculated by Student's *t*-test. Data are expressed as mean ± standard error of the mean (SEM).

|  | Whole brain | |
| --- | --- | --- |
|  | **WT** | **CD** |
| 2-AG (nmol/g) | 4.96 ± 0.08 | 5.51 ± 0.29 |
| 2-LG (nmol/g) | 0.42 ± 0.03 | 0.46 ± 0.03 |
| 2-OG (nmol/g) | 0.96 ± 0.04 | 0.96 ± 0.05 |
| AEA (pmol/g) | 5.57 ± 0.19 | 5.65 ± 0.32 |
| DEA (pmol/g) | 1.98 ± 0.07 | 1.93 ± 0.06 |
| DHEA (pmol/g) | 12.38 ± 0.35 | 11.63 ± 0.60 |

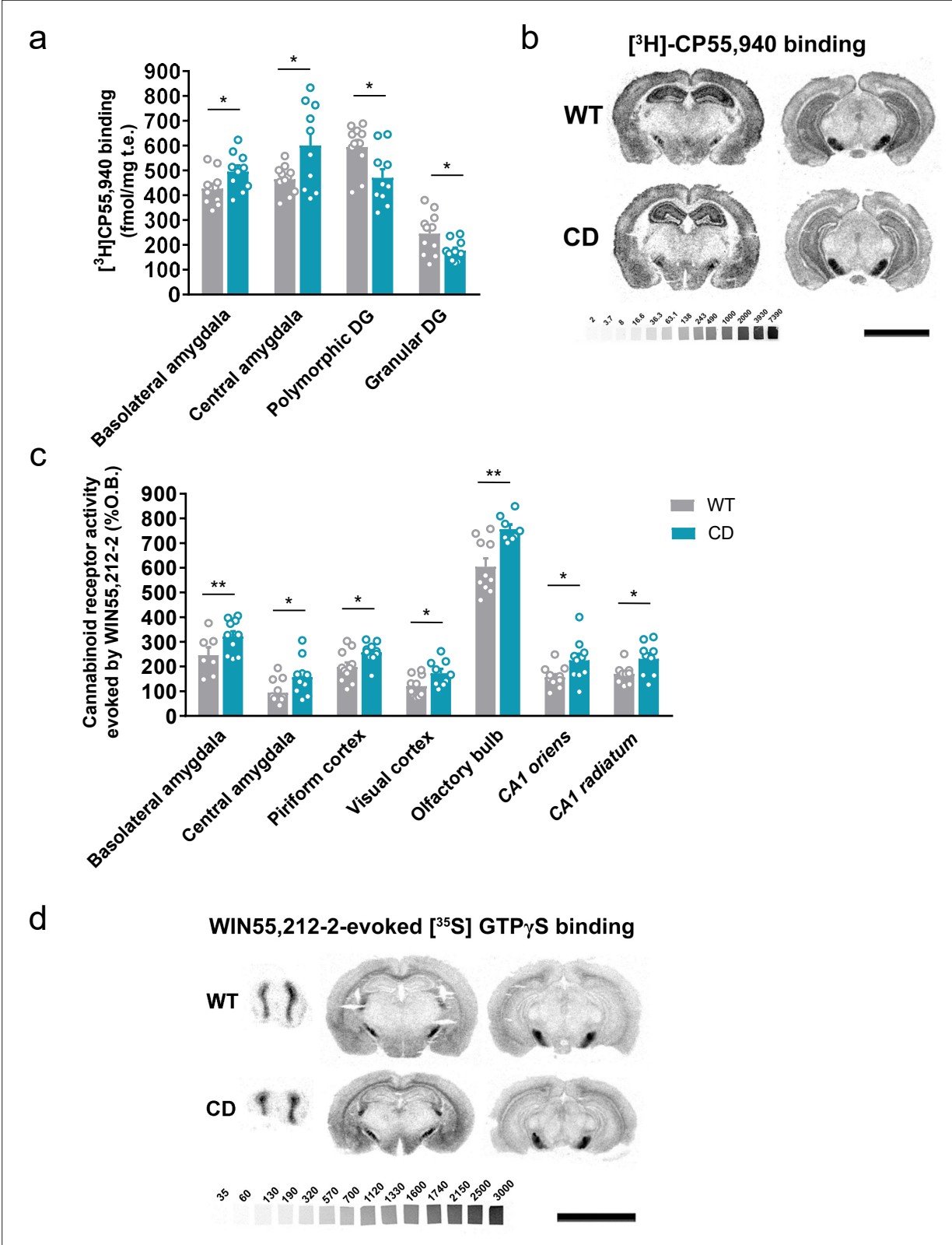

**Figure 2.** Complete deletion (CD) mice show alterations in cannabinoid type-1 receptor (CB1R) density and coupling to $G_{i/o}$ proteins. (**a**) [³H]CP55,940 binding of brain regions with significant changes in CD mice in comparison to WT littermates (WT, $n = 11$; CD, $n = 10$). (**b**) Representative images of [³H] CP55,940-binding autoradiography. (**c**) Brain regions showing significant changes in [³⁵S]GTPγS binding evoked by WIN55,212–2 (10 µM) in CD mice in comparison to WT littermates (WT, $n = 10$–11; CD, $n = 8$–10), expressed as percentage of stimulation over the basal binding. (**d**) Representative images

*Figure 2 continued on next page*

*Figure 2 continued*

of WIN55,212–2-evoked [$^{35}$S]GTPγS binding. [$^{14}$C]-microscales used as standards in Ci/g t.e. Scale bar = 5 mm. Statistical significance was calculated by Student's *t*-test. *$p < 0.05$; **$p < 0.01$; (genotype effect). Data are expressed as mean ± standard error of the mean (SEM).

The online version of this article includes the following source data and figure supplement(s) for figure 2:

**Source data 1.** Cannabinoid type-1 receptor (CB1R) density and coupling to G$_{i/o}$ proteins in complete deletion (CD) mice.

**Figure supplement 1.** Complete deletion (CD) mice alterations in cannabinoid receptor density and activity are specific for cannabinoid type-1 receptor (CB1R).

subtype of the cannabinoid receptor studied, SR141716A (rimonabant) and SR144528, selective antagonists for CB1R and CB2R, respectively, were used. SR141716A, but not SR144528, blocked [$^3$H] CP55,940 radioligand binding in brain slices (*Figure 2—figure supplement 1a*) indicating that the observed changes occur in CB1R distribution.

Next, we assessed cannabinoid receptor-mediated G$_{i/o}$ protein activity by [$^{35}$S]GTPγS autoradiography. CD mice exhibited a higher WIN55,212–2-stimulated [$^{35}$S]GTPγS binding in several brain regions compared to WT mice, while basal activity was similar in both genotypes. The regions in CD mice with high G-protein coupling included the basolateral and central amygdala, piriform and visual cortex, olfactory bulb, and *CA1 stratum oriens* and *CA1 stratum radiatum* (*Figure 2c, d*; *Supplementary file 2*). This increase was blocked in the presence of the CB1R antagonist SR141716A, but not with the CB2 antagonist SR144528 (*Figure 2—figure supplement 1b*). These results indicated that there was an increase in the functional coupling of CB1R to G$_{i/o}$ proteins in several brain regions of CD mice. Interestingly, both CB1R density and CB1R-mediated G$_{i/o}$ protein activity increased in different subregions of amygdala.

## JZL184 administration corrects behavioral impairment in CD mice

Previous reports have demonstrated that subchronic administration of JZL184, an irreversible inhibitor of the MAGL, promotes downregulation and G-protein uncoupling of CB1R (*Llorente-Ovejero et al., 2018*; *Kinsey et al., 2013*). We found that administration of JZL184 (8 mg/kg, i.p.) for 10 days significantly decreased the time that CD mice spent exploring unfamiliar juvenile S1, reaching levels comparable to those displayed by WT mice during the sociability phase (*Figure 3a*, center), whereas no significant changes were observed after a single dose of the drug (*Figure 3—figure supplement 1*). During the preference for social novelty phase, CD mice treated with JZL184 showed a preference for unfamiliar juvenile S2 similar to WT animals (*Figure 3a*, right). Notably, administration of JZL184 did not alter the exploration time of WT mice or the exploration time during the habituation phase (*Figure 3a*, right). Furthermore, no changes were observed in locomotor activity after JZL184 administration in WT or in CD mice (*Figure 3—figure supplement 2*) discarding an overall effect of treatment that could bias exploratory activity of mice.

Given the role of the ECS in learning and memory processes (*Marsicano and Lafenêtre, 2009*), we studied the effect of JZL184 treatment over the short-term recognition memory deficit of CD mice. Subchronic administration of JZL184 (8 mg/kg, i.p.) for 7 days (last administration 2 hr before starting the training phase of the NORT), restored memory impairment in CD mice (*Figure 3b*).

Then, we assessed anxiety-like behavior using the elevated plus maze test after 10 days of treatment. We observed that CD mice spent more time in the open arms of the maze than WT animals (*Figure 3c*, left). Interestingly, this behavior of CD mice was reversed after JZL184 treatment. However, the number of total entries, as a measure of exploratory activity in the elevated plus maze, showed much variability without revealing any significant effect (*Figure 3c*, right).

## JZL184 administration modifies ECS signaling in CD mice

We first confirmed the efficacy of JZL184 enhancing 2-AG levels in the brain by assessing endocannabinoid levels in two of the main regions of interest, amygdala and hippocampus. JZL184 treatment increased 2-AG levels in both brain regions and showed similar efficacy in WT and CD mice. No changes were found in AEA levels among groups (*Table 2*).

We focused on the amygdala to determine whether subchronic administration of JZL184 at 8 mg/kg for 10 days induced changes in CB1R density and CB1 receptor-mediated G$_{i/o}$ protein activity. We found by immunofluorescence analysis of basolateral amygdala an enhanced expression of CB1R in

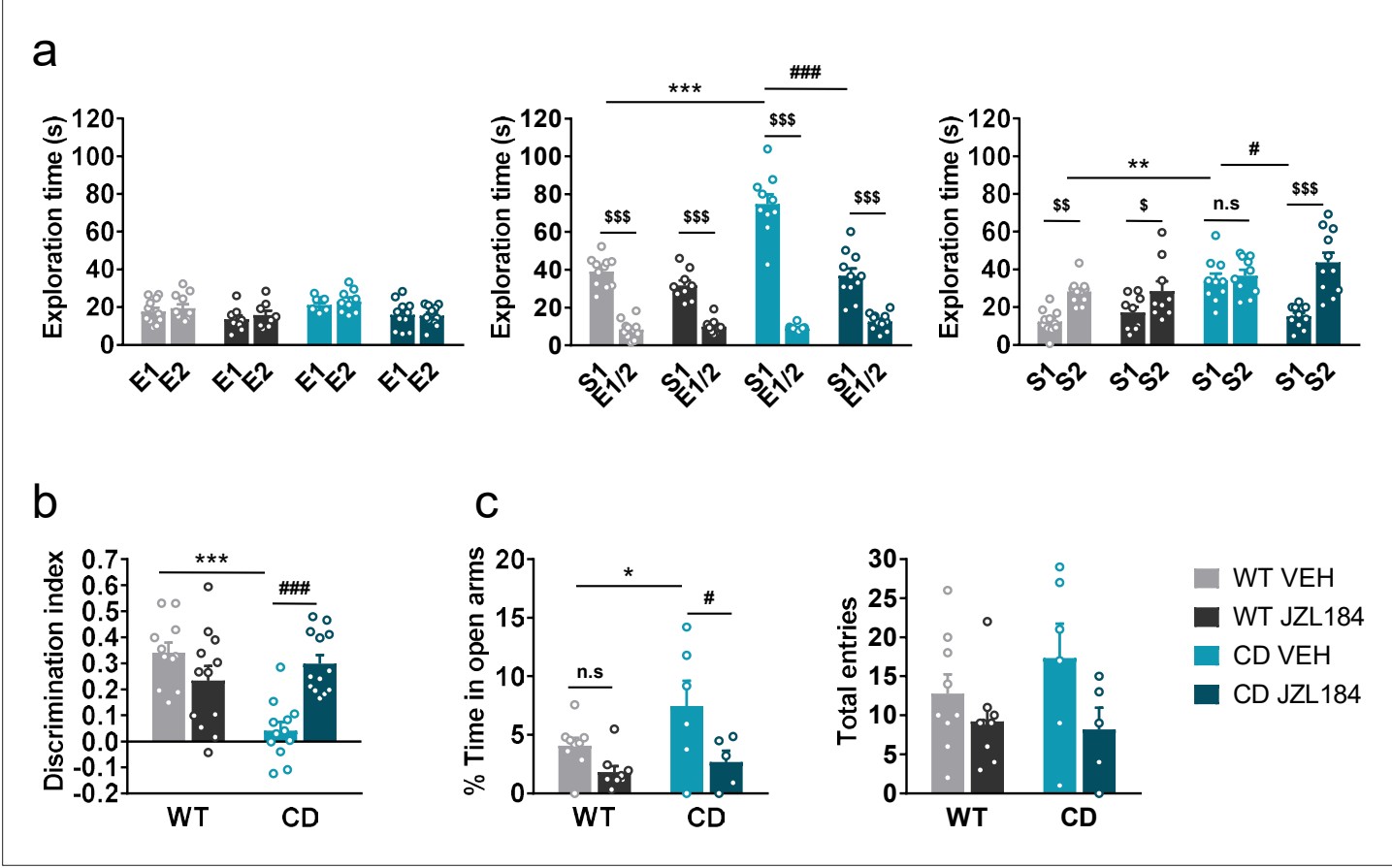

**Figure 3.** JZL184 treatment normalizes behavioral traits of complete deletion (CD) mice. (**a**) Time spent exploring either empty compartments (E) or stranger mice (S) in the Vsocial-maze after 10 days of treatment with vehicle (VEH) or JZL184 (8 mg/kg) (WT VEH, $n$ = 11; WT JZL184, $n$ = 8–9; CD VEH, $n$ = 9–10; CD JZL184, $n$ = 11). Statistical significance was calculated by repeated measures analysis of variance (ANOVA) comparison. $^\$$p < 0.05; $^{\$\$}$p < 0.01; $^{\$\$\$}$p < 0.001 (compartment effect); *p < 0.05; ***p < 0.001 (genotype effect); #p < 0.05; ###p < 0.001 (treatment effect). (**b**) Discrimination index of WT and CD mice treated for 7 days with vehicle (VEH) or JZL184 (8 mg/kg) (WT VEH, $n$ = 11; WT JZL184, $n$ = 12; CD VEH, $n$ = 12; CD JZL184, $n$ = 13). Statistical significance was calculated by Newman–Keuls post hoc test following two-way ANOVA. ***p < 0.001 (genotype effect); ###p < 0.001 (treatment effect). (**c**) Percentage of time spend in open arms and total entries in the elevated plus maze of WT and CD mice treated for 10 days with vehicle (VEH) or JZL184 (8 mg/kg) (WT VEH, $n$ = 9; WT JZL184, $n$ = 9; CD VEH, $n$ = 6; CD JZL184, $n$ = 5). Statistical significance was calculated by Newman–Keuls post hoc test following two-way ANOVA. *p < 0.05 (genotype effect); #p < 0.05 (treatment effect). Data are expressed as mean ± standard error of the mean (SEM).

The online version of this article includes the following source data and figure supplement(s) for figure 3:

**Source data 1.** JZL184 treatment normalizes behavioral traits of complete deletion (CD) mice.

**Figure supplement 1.** Acute administration of JZL184.

**Figure supplement 1—source data 1.** Acute administration of JZL184.

**Figure supplement 2.** Locomotor activity after JZL184 treatment.

**Figure supplement 2—source data 1.** Locomotor activity after JZL184 treatment.

CD mice that was significantly decreased after JZL184 treatment (*Figure 4a, b*). Similarly, a downregulation of CB1R was observed by immunoblot in amygdala homogenates from CD mice treated with JZL184 (*Figure 4—figure supplement 1*). Such modulation in CB1R was not paralleled by changes in the density of cells positive for the neuronal marker Neu N (*Figure 4—figure supplement 2a*) or the density of cells stained with 4',6-diamidino-2-phenylindole (DAPI) (*Figure 4—figure supplement 2b*), discarding major contributions of such confounding factors in CB1R changes in expression.

We then analyzed CB1R-mediated $G_{i/o}$ protein activity in the brain regions where we previously found changes in CD mice. Interestingly, significant changes were observed after JZL184 treatment in functional coupling of CB1R to $G_{i/o}$ proteins in three brain regions of CD mice: basolateral amygdala, olfactory bulb, and CA1 *stratum radiatum* (*Figure 4c, d*). Curiously, these changes in CB1R activity

**Table 2.** Levels of the two major endocannabinoids (2-AG and AEA) in brain regions relevant for social and cognitive behavior (amygdala and hippocampus) after 10 days treatment with VEH or JZL184.

(WT VEH, *n* = 6; WT JZL184, *n* = 6; CD VEH, *n* = 7; CD JZL184, *n* = 7). Statistical significance was calculated by Newman–Keuls post hoc test following two-way analysis of variance (ANOVA). ###$p <$ 0.001 (treatment effect). Data are expressed as mean ± standard error of the mean (SEM).

| | Amygdala | | | |
|---|---|---|---|---|
| | **WT VEH** | **WT JZL184** | **CD VEH** | **CD JZL184** |
| 2-AG (nmol/g) | 18.72 ± 4.60 | 122.96 ± 21.99### | 16.99 ± 0.86 | 128.33 ± 18.12### |
| AEA(pmol/g) | 6.89 ± 1.04 | 5.34 ± 0.53 | 6.64 ± 0.41 | 4.69 ± 0.41 |
| | **Hippocampus** | | | |
| | **WT VEH** | **WT JZL184** | **CD VEH** | **CD JZL184** |
| 2-AG (nmol/g) | 9.11 ± 1.45 | 75.55 ± 14.95### | 9.01 ± 0.46 | 86.46 ± 15.25### |
| AEA (pmol/g) | 8.30 ± 0.77 | 8.26 ± 0.57 | 8.80 ± 0.63 | 7.85 ± 0.87 |

were not observed in other areas such as the central amygdala, the pyriform cortex, visual cortex or CA1 *stratum oriens* (*Figure 4c, d*) further supporting local effects of JZL184 treatment. These data demonstrated that the subchronic treatment of JZL184 normalizes alterations in CB1R density and activity in key brain regions of CD mice.

## JZL184 treatment has an impact on the cardiovascular phenotype of CD mice

We hypothesized that JZL184 treatment could impact the cardiovascular phenotype of CD mice. For this purpose, several anatomical and functional parameters were evaluated. In agreement with previous descriptions of the CD model, mice presented an increase in the heart weight/body weight ratio (*Figure 5a*) and a nonsignificant trend on muscle proportion of the left ventricle (*Figure 5— figure supplement 1*) consistent with cardiac hypertrophy. These parameters were not paralleled by other measures such the brain weight/body weight ratio (*Figure 5—figure supplement 2*). Notably, administration of JZL184 for 10 days (8 mg/kg, i.p.) restored the heart weight/body weight (*Figure 5a*) but did not modify brain size parameters (*Figure 5—figure supplement 2*), suggesting a specific improvement over cardiac hypertrophy of CD mice. To confirm the results on the hypertrophic pheno-type, we measured the cross-sectional area (CSA) of cardiomyocytes in the ventricle by histological methods. CSA was increased in CD mice and was completely normalized after JZL184 treatment, reaching the level of WT animals further validating the beneficial effect of the pharmacological inter-vention over cardiac hypertrophy (*Figure 5b, c*) without any significant effect on WT mice. To further confirm these changes, we performed transthoracic echocardiography. According to echocardiog-raphy measurements, left ventricular mass and wall thickness at the interventricular septum (IVS) and the posterior wall (LVPW) were significantly increased in CD mice and there was a tendency toward a decrease after JZL184 treatment (*Supplementary file 3*). In addition, echocardiography analysis revealed that CD mice presented a slight reduction in left ventricular ejection fraction (LVEF) at base-line that was normalized after JZL184 (*Figure 5d*), indicating that the treatment produced an overall improvement in cardiac function.

We additionally measured systolic blood pressure and found that JZL184 treatment prevented the mild hypertension described in CD mice (*Segura-Puimedon et al., 2014*) without affecting blood pressure in WT mice (*Figure 5e*).

To further explore the mechanisms behind, we assessed CB1R mRNA expression levels in heart homogenate samples. Notably, we found an increase in the expression of *Cnr1* in CD mice that was reversed after the subchronic administration of JZL184 (*Figure 5f*).

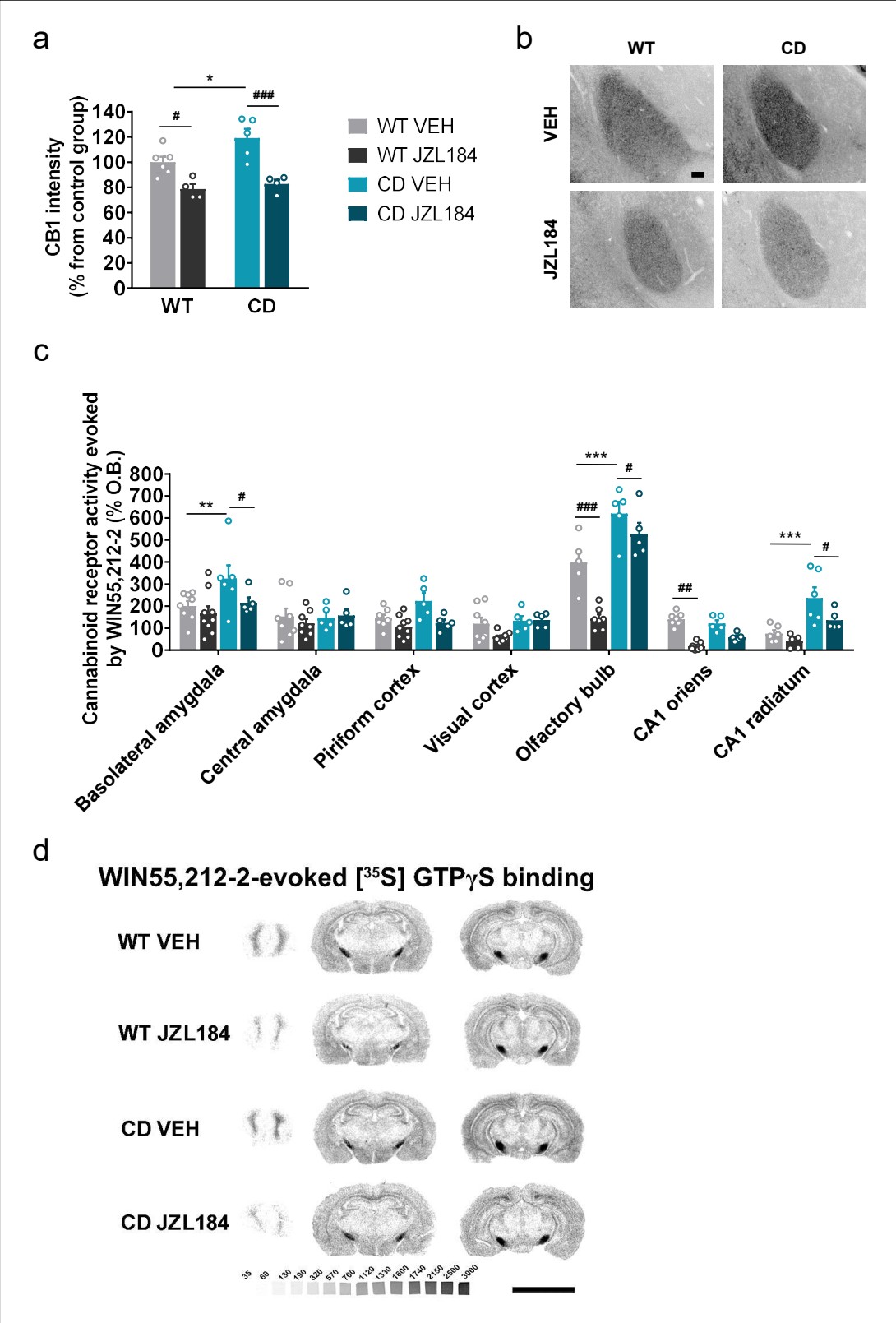

**Figure 4.** JZL184 treatment restores altered cannabinoid type-1 receptor (CB1R) density and coupling to $G_{i/o}$ proteins in complete deletion (CD) mice. (**a**) Quantification and (**b**) representative images of CB1R immunodetection in the basolateral amygdala of WT and CD mice after 10 days of treatment with vehicle (VEH) or JZL184 (8 mg/kg) (WT VEH, $n = 6$; WT JZL184, $n = 4$; CD VEH, $n = 5$; CD JZL184, $n = 4$). Scale bar = 100 μm. Statistical significance was calculated by Newman–Keuls post hoc test following two-way analysis of variance (ANOVA). *$p < 0.05$ (genotype effect); #$p < 0.05$, ###$p < 0.001$

*Figure 4 continued on next page*

*Figure 4 continued*

(treatment effect). (**c**) [$^{35}$S]GTPγS binding evoked by WIN55,212–2 (10 µM) after 10 days of treatment with vehicle (VEH) or JZL184 (8 mg/kg) (WT VEH, *n* = 5–8; WT JZL184, *n* = 6–9; CD VEH, *n* = 5–6; CD JZL184, *n* = 5) expressed as percentage of stimulation over the basal binding. Statistical significance was calculated by Newman–Keuls post hoc test following two-way ANOVA. **p < 0.01, ***p < 0.001 (genotype effect); #p < 0.05, ##p < 0.01, ###p < 0.001 (treatment effect). (**d**) Representative images of WIN55,212–2-evoked [$^{35}$S]GTPγS binding. [$^{14}$C]-microscales used as standards in Ci/g t.e. Scale bar = 5 mm. Data are expressed as mean ± standard error of the mean (SEM).

The online version of this article includes the following source data and figure supplement(s) for figure 4:

**Source data 1.** JZL184 treatment restores altered cannabinoid type-1 receptor (CB1R) density and coupling.

**Figure supplement 1.** JZL184 treatment downregulates cannabinoid type-1 receptor (CB1R) protein levels in the amygdala of complete deletion (CD) mice (CD VEH, n=7; CD JZL184, n=7).

**Figure supplement 1—source data 1.** JZL184 treatment downregulates cannabinoid type-1 receptor (CB1R) protein levels in the amygdala of complete deletion (CD) mice.

**Figure supplement 1—source data 2.** Original blots of cannabinoid type-1 receptor (CB1R) in the amygdala of complete deletion (CD) mice.

**Figure supplement 2.** Quantification of total number of cells and neurons in basolateral amygdala.

**Figure supplement 2—source data 1.** Quantification of total number of cells and neurons in basolateral amygdala.

## JZL184 treatment reverses cardiac transcriptional deficits of CD mice

Given the results on the cardiac phenotype in the WBS mouse model, we performed a transcriptomic analysis on cardiac tissue to further explore the effects of JZL184 treatment in CD mice. For this purpose, we performed high-throughput RNA sequencing (RNA-seq) of heart samples from mice treated with vehicle or JZL184 for 10 days. Before analyzing differences in gene expression, a principal component analysis was performed that revealed the samples used were informative with respect to the differences between experimental groups (*Figure 6—figure supplement 1*). After that, we first compared RNA-seq data between WT and CD treated with vehicle, and found 3838 differentially expressed genes (DEGs) with a |log$_2$ fold-change| > 0 and adjusted p values <0.05 excluding the genes of the WBS critical region (*Figure 6a*). Of these DEGs, 1882 were upregulated and 1956 down-regulated, indicating a relevant alteration in cardiac gene expression in CD mice. Enrichment analysis identified Gene Ontology (GO) biological processes linked to the cardiovascular system including striated muscle cell development, muscle cell development, muscle system process, cardiac muscle contraction, heart contraction, heart process, contraction muscle cell development, and muscle tissue progress among others (*Figure 6b*). Then, we calculated the DEGs comparing vehicle- and JZL184-treated CD mice. This yielded 2122 upregulated and 1990 downregulated genes as a result of the treatment (*Figure 6c*). Interestingly, overlap analysis revealed that 1433 DEGs, 73% of total downregulated in CD vehicle, were upregulated following JZL184 treatment (*Figure 6d*). Enrichment analysis identified significant changes in several GO biological processes. Notably, among the 10 most significant GO biological processes, the majority were related to cardiovascular function including heart contraction, heart process, regulation of heart contraction, muscle system process, cardiac muscle contraction, muscle cell development, and regulation of blood circulation (*Figure 6d*). In addition, 1262 DEGs, 67% of total upregulated in CD vehicle, were downregulated after JZL184 treatment. Enrichment analysis identified much more diverse GO biological processes than those observed on the set of DEGs upregulated after JZL184 treatment, but also included some GO biological processes linked to the cardiovascular system such as regulation of vasculature development, endothelial cell migration, and blood vessel endothelial cell migration (*Figure 6d*). We put together a set of relevant pathways associated to cardiac function and hypertrophy to assess in a targeted manner the implication in CD mice and the effect of JZL184 (*Supplementary file 4*). We observed many of the genes differentially expressed in CD mice related with these selected pathways. Surprisingly, the expression of most altered genes associated to cardiac function and hypertrophy in CD mice was reverted after JZL184 treatment and therefore could had a significant beneficial effect toward the pathological condition (*Supplementary file 4*). Moreover, when cardiovascular genes were only considered, the proportion of overlapping genes that reversed their expression as a result of treatment was higher than that described above: 79% of downregulated and 69% of upregulated in CD vehicle. Moreover, a concordance was observed in the direction and magnitude of the change in reverting cardiovascular genes (Pearson *R* = −0.9666, p value = 2.2e−16) (*Figure 6e*). Overall, these data indicated that

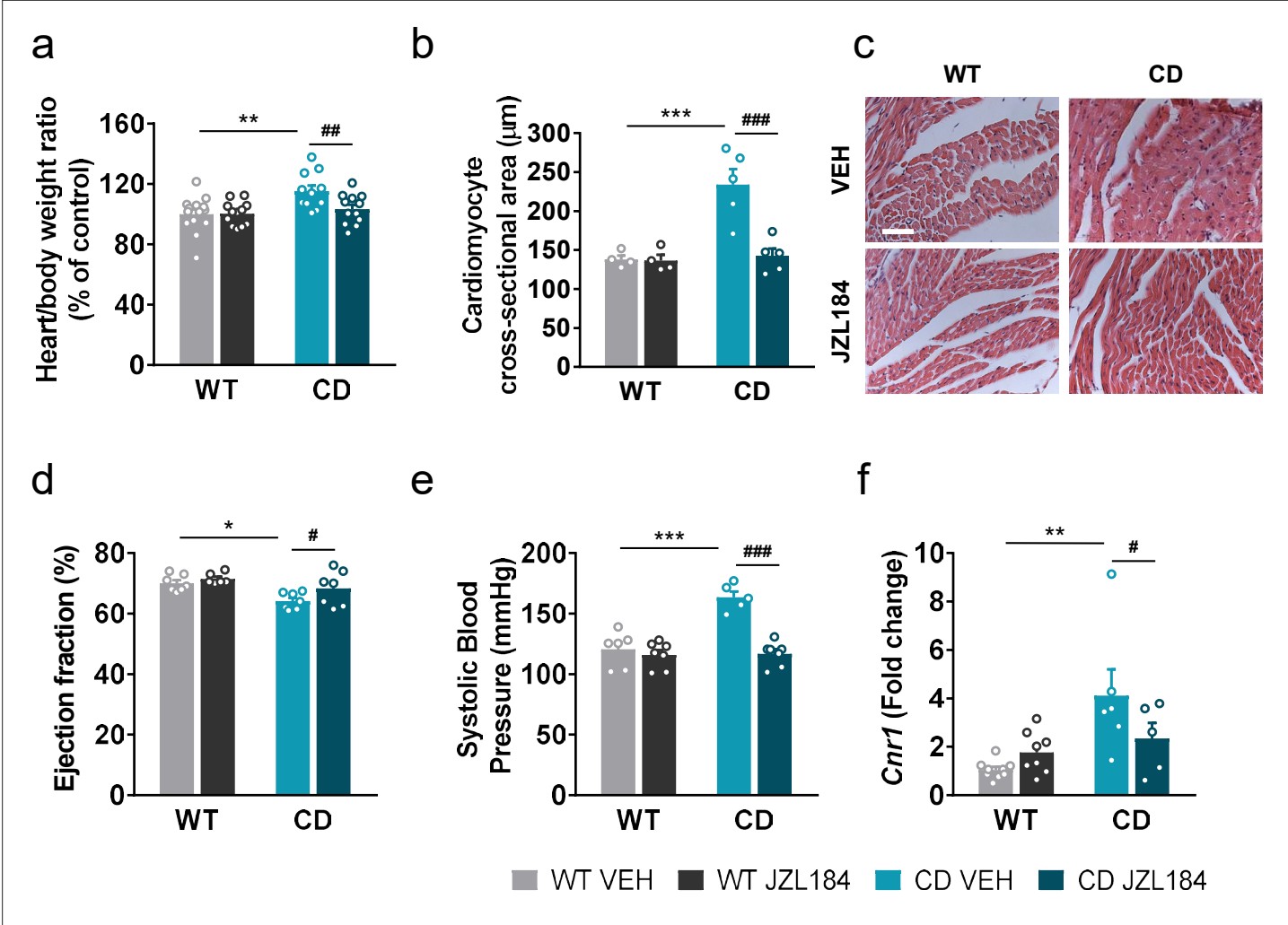

**Figure 5.** JZL184 administration regresses cardiac hypertrophy and the expression of cardiac *Cnr1* alterations of complete deletion (CD) mice. (**a**) Heart/body weight ratios obtained from WT and CD mice treated for 10 days with vehicle (VEH) or JZL184 (8 mg/kg) (WT VEH, *n* = 14; WT JZL184, *n* = 12; CD VEH, *n* = 11; CD JZL184, *n* = 12). (**b**) Cardiomyocyte cross-sectional area measured from WT and CD mice after treatment (WT VEH, *n* = 4; WT JZL184, *n* = 4; CD VEH, *n* = 5; CD JZL184, *n* = 5) and (**c**) representative images, scale bar = 5 μm. (**d**) Ejection fraction (%) assessed by echocardiography from measurements performed on bidimensional images (WT VEH, *n* = 7; WT JZL184, *n* = 6; CD VEH, *n* = 7; CD JZL184, *n* = 7). (**e**) Systolic blood pressure (mmHg) obtained after 10 days treatment (WT VEH, *n* = 6; WT JZL184, *n* = 7; CD VEH, *n* = 5; CD JZL184, *n* = 7) (**f**) Cardiac mRNA levels of *Cnr1* gene obtained by qPCR expressed in fold-change after the 10th day administration (WT VEH, *n* = 9; WT JZL184, *n* = 8; CD VEH, *n* = 6; CD JZL184, *n* = 5). Statistical significance was calculated by Newman–Keuls post hoc test following two-way analysis of variance (ANOVA). *$p < 0.05$; **$p < 0.01$; ***$p < 0.001$ (genotype effect); #$p < 0.05$; ##$p < 0.01$; ###$p < 0.001$ (treatment effect). Data are expressed as mean ± standard error of the mean (SEM).

The online version of this article includes the following source data and figure supplement(s) for figure 5:

**Source data 1.** JZL184 treatment has an impact on the cardiovascular phenotype of complete deletion (CD) mice.

**Figure supplement 1.** Additional heart morphology data.

**Figure supplement 1—source data 1.** Additional heart morphology data.

**Figure supplement 2.** Brain weight measurements.

**Figure supplement 2—source data 1.** Brain weight measurements.

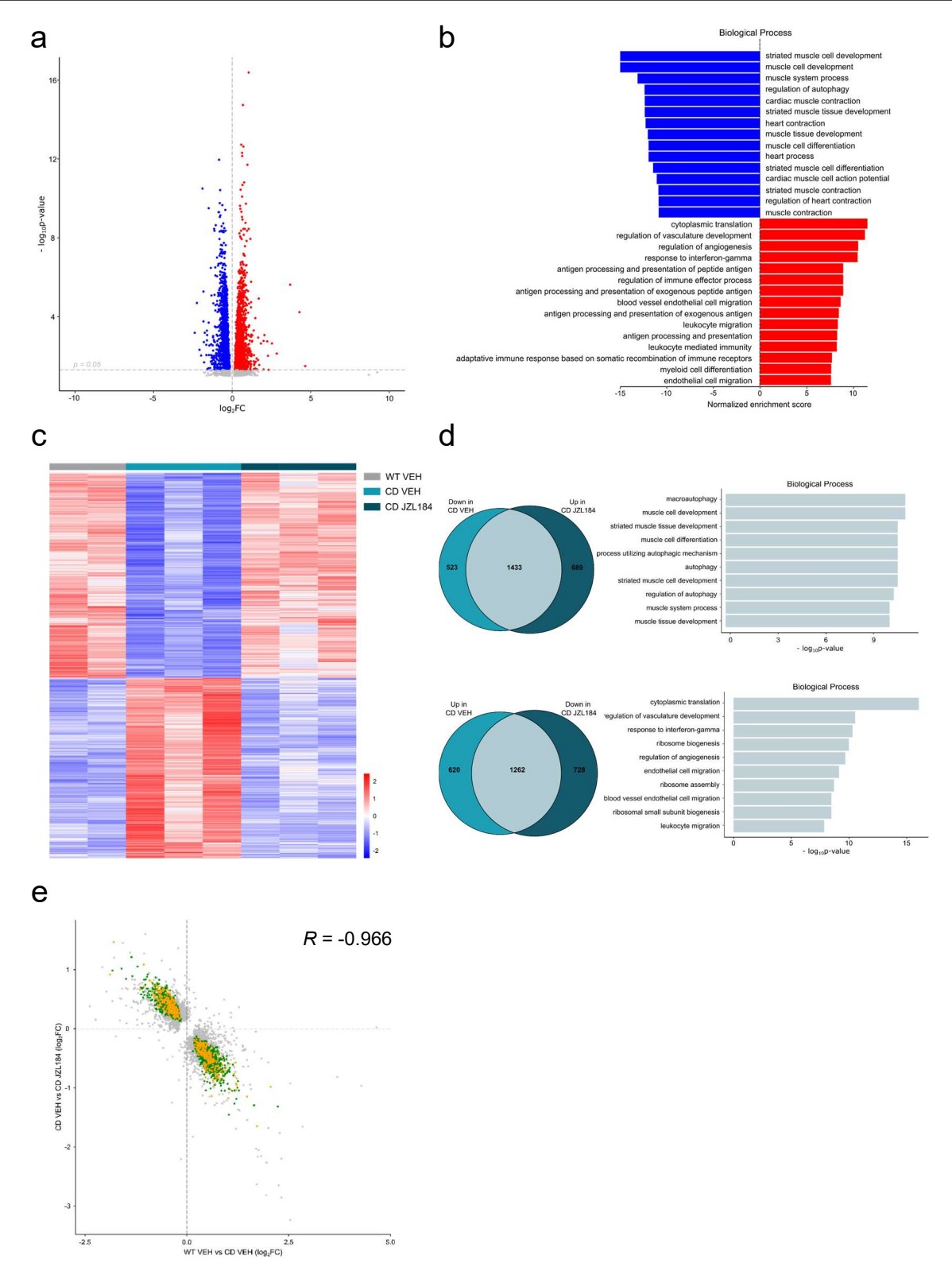

**Figure 6.** JZL184 treatment reversed alterations of the cardiac transcriptome observed in complete deletion (CD) mice. (**a**) Volcano plot of differentially expressed genes (p < 0.05 and |log$_2$FC| > 0) between CD and WT mice. Red indicates relative increased expression and blue indicates relative decreased expression. (**b**) Gene ontology enrichment analysis for both up- and downregulated genes in CD mice compared with WT. Most significant biological processes terms are represented for each group. (**c**) Heatmap showing the relative mRNA expression level of genes that reverted their

*Figure 6 continued on next page*

*Figure 6 continued*

expression in CD mice treated for 10 days with JZL184 (8 mg/kg) and CD or WT littermates treated with vehicle. (**d**) Venn diagrams and gene ontology enrichment analysis of genes that showed opposite differential expression in CD mice after 10 days treatment with JZL184 (8 mg/kg) compared with CD mice treated with vehicle. (**e**) Correlation plot of differentially expressed genes in CD mice treated for 10 days with JZL184 (8 mg/kg) and CD or WT littermates treated with vehicle. In green, genes with opposite differential expression between conditions, in orange reverted genes associated with cardiovascular function, in gray genes with no change (WT VEH, $n = 2$; CD VEH, $n = 3$; CD JZL184, $n = 3$).

The online version of this article includes the following source data and figure supplement(s) for figure 6:

**Source data 1.** JZL184 treatment reverses cardiac transcriptional deficits of complete deletion (CD) mice.

**Figure supplement 1.** Principal component analysis (PCA).

**Figure supplement 1—source data 1.** Principal component analysis (PCA).

JZL184 treatment may restore the normal expression of genes relevant for cardiovascular function that were either down- or upregulated in CD mice.

## Discussion

The results in this study show that the ECS can be used as a target to improve characteristic behavioral and cardiac phenotypes in a relevant mouse model for WBS.

We first assessed social behavior in CD mice using an approach that allowed us to assess both sociability and preference for social novelty. In line with previous findings (*Segura-Puimedon et al., 2014*), we observed a significant increase in the sociability of these mice compared to WT mice. This phenotype resembles the human condition in which WBS subjects show higher social motivation (*Riby et al., 2013*; *Riby and Hancock, 2009*). In addition, we described for the first time that CD mice did not show preference for social novelty. This lack of social novelty preference is dependent on social stimuli since CD mice assessed on object novelty behaved similar to WT mice, and reflect a lack of habituation of CD mice to the previously encountered stranger mouse (*Segura-Puimedon et al., 2014*). This trait is reminiscent of the lack of habituation to faces observed through electrodermal measures in WBS individuals, which may cause that social stimuli appear continuously novel and interesting (*Järvinen et al., 2012*). These results improved the characterization of the social behavior of CD mice as a significant model of WBS.

The analysis of the main components of the ECS revealed an increased density and functional coupling to $G_{i/o}$ proteins of CB1R in specific brain regions. Changes in CB1R density in polymorphic and granular layers of dentate gyrus were not accompanied by a commensurate decrease in CB1R-mediated $G_{i/o}$ protein activity suggesting a compensatory increase of CB1R function in these regions. Notably, density and functional coupling were increased in the amygdala, a region with major role in social behavior and where structural and functional abnormalities have been described in WBS patients (*Meyer-Lindenberg et al., 2005*; *Muñoz et al., 2010*; *Capitão et al., 2011*).

Subchronic administration of the compound JZL184 at 8 mg/kg reduced to WT levels the time that CD mice spent exploring a stranger mouse. In addition, CD-treated mice spent more time exploring an unfamiliar stranger mouse instead of a familiar one, which indicates a restoration of the preference for social novelty and, it may suggest an improvement in the lack of habituation to social stimuli of CD mice. The treatment did not have any effect over WT animals, revealing that the effects of JZL184 over social interactions were selective for the social phenotype of CD mice.

In the context of social behavior, previous results using JZL184 showed that a single systemic dose of this compound increased social play in juvenile rats (*Manduca et al., 2016*) and social interaction in a mouse model of ASD, Shank3B−/− (*Folkes et al., 2020*), and repeated social stress enhanced brain levels of 2-AG as well as decreased CB1R density (*Tomas-Roig et al., 2016*). We did not observe a major effect over social behavior after a single dose of JZL184 in CD mice suggesting that changes in CB1R density and signaling after chronic exposure may be determinants of the effect observed over social behavior in CD mice.

Together, we demonstrated that the subchronic administration of JZL184 at 8 mg/kg normalizes social abnormalities of CD mouse model that resemble the human condition. Atypical social functioning of WBS subjects predisposes to social vulnerability (*Jawaid et al., 2012*). In fact, WBS subjects have difficulties in peer interactions, maintaining friendships, and around 73% have experienced social isolation (*Davies et al., 1998*). In addition, they have an increased risk to suffer psychiatric conditions

that are not associated to the IQ range or to language disability (*Stinton et al., 2010*) and seem related to the hypersocial phenotype (*Riby et al., 2014*; *Ng-Cordell et al., 2018*). Therefore, improvements in social functioning may have a beneficial effect over the quality of life of WBS subjects.

The treatment with the MAGL inhibitor JZL184 also restored short-term hippocampal-memory deficits of CD mice. In line with previous findings, our data also revealed that the JZL184 treatment did not have any effect over WT mice memory performance (*Busquets-Garcia et al., 2011*) and therefore, it is specific for the disorder context. Consequentially, our results on memory performance after JZL184 treatment could be further explored for cognitive restoration in the context of WBS.

As expected, we observed an increase in 2-AG brain levels resulting from JZL184 treatment for 10 consecutive days. In addition, we showed a decrease in CB1R density in the basolateral amygdala of CD mice. These results agree with previous studies that have pointed to a downregulation of CB1R in chronic JZL184-exposed animals (*Llorente-Ovejero et al., 2018*; *Kinsey et al., 2013*; *Schlosburg et al., 2010*). Moreover, we observed a normalization of functional coupling to $G_{i/o}$ proteins of CB1R in the basolateral amygdala, the olfactory bulb and the *CA1 stratum radiatum* of CD mice. These results indicate that treatment with JZL184 is normalizing alterations in CB1R signaling in different key brain regions of CD mice, and that these may be mediating the beneficial effect of the treatment on social novelty and cognition.

Notably, treatment with JZL184 may have additional benefits in WBS since it proved to have positive effects on the cardiac phenotype of CD mice. CD animals present a cardiovascular phenotype with cardiac hypertrophy present from an early age and accompanied by a decrease in the ejection fraction. We observed a normalization of the heart weight/body weight ratio and left ventricular wall thickness and size of cardiomyocytes after treatment with JZL184, indicating an improvement in heart hypertrophy. Moreover, ejection fraction was also improved after JZL184 treatment. In addition, mild hypertension observed in CD mice was also reversed after JZL184 treatment. This could be of major clinical relevance, given that the cardiovascular phenotype is the most life-threatening complication of the disorder, and constitutes a relevant new potential therapeutic approach for cardiovascular disorders.

CB1R are predominantly located in the cardiovascular centers of the brainstem and hypothalamus but also in myocardium, postganglionic autonomic nerve terminals, and vascular endothelial and smooth muscle cells. Therefore, modulation of the ECS may have effects on the cardiovascular system by multiple mechanisms (*Haspula and Clark, 2020*). Both agonists and antagonists of CB1R have shown to be beneficial for cardiovascular function in mouse models of different conditions (*Bátkai et al., 2004*; *Liao et al., 2013*; *Slavic et al., 2013*) pointing that effects of the modulation of CB1R over cardiovascular systems seems to be highly dependent on the context (*Haspula and Clark, 2020*). Several studies have demonstrated that antihypertensive drugs are known to cause a regression of left ventricular hypertrophic phenotype (*van Zwieten, 2000*). Therefore, a decrease in blood pressure in CD mice may play a role in the normalization of the hypertrophic phenotype. In addition, we observed the upregulation of CB1R mRNA levels of heart homogenates of CD mice in comparison to WT mice, while normalized after JZL184 treatment, which could be related to the beneficial effect of the drug.

Our transcriptome analysis of the heart tissue revealed that expression of the majority of genes that were down- or upregulated in the CD mice, and were linked to cardiovascular function (including cardiac muscle contraction and development, and endothelial cell migration and vasculature development), were brought back to control levels after treatment with JZL184. Of special significance was the genes deregulated in hypertrophic phenotypes, which were surprisingly responsive to the JZL184 treatment in the CD context. We also found an enrichment of genes related to macroautophagy/autophagy in CD samples after JZL184 treatment. These biological processes have been previously associated with improved cardiac function after heart failure (*Yamaguchi, 2019*). We also found a decrease in expression of genes involved in translation and ribosomal function, which could be associated with reduced hypertrophy (*Hannan et al., 2003*). Further experiments may address the status of the different components of the ECS in the WBS context to better understand the mechanism of action of JZL184 over the cardiovascular system.

Additionally, other strategies of modulation such as inhibitors of CB1R activity could be worth assessing in CD mice, since one marked effect of repeated JZL184 was to desensitize CB1R in specific regions of the brain. Approaches with inhibitors could also help understand whether the effects of JZL184 are mainly due to the inhibition of CB1R function, or to other targets of enhanced 2-AG levels.

Taken together, the results of this study are of great importance given the few preclinical studies addressing potential treatments for WBS. In this regard, the modulation of the ECS may be an appropriate novel therapeutic strategy to tackle not only the social phenotype but also memory shortfalls and cardiovascular deficits in WBS.

## Materials and methods

**Key resources table**

| Reagent type (species) or resource | Designation | Source or reference | Identifiers | Additional information |
|---|---|---|---|---|
| Strain, strain background (*Mus musculus*, male) | C57BL/6J | Charles Rivers, France | C57Bl/6J | Male |
| Genetic reagent (*Mus musculus*) | CD | University Pompeu Fabra | | (C57BL/6J background, male) |
| Chemical compound, drug | JZL184 | Abcam | ab141592 | |
| Chemical compound, drug | Dimethyl sulfoxide | Scharlau Chemie | SU01531000 | |
| Chemical compound, drug | Polyethylene glycol 400 | AppliChem | 142436.1611 | |
| Chemical compound, drug | Tween-80 | Sigma-Aldrich | P1754 | |
| Chemical compound, drug | 0.9%, NaCl physiological saline | Laboratorios Ern | Vitulia | |
| Chemical compound, drug | Ketamidor | Richter pharma | K1NAI027 | |
| Chemical compound, drug | Xylazine hydrochloride | Sigma-Aldrich | X1251 | |
| Biological sample (*Equus asinus*) | Normal donkey serum | Sigma-Aldrich | D9663-10ML | 3% in PBS with 0.3% Triton X-100 |
| Antibody | Anti-CB1R (rabbit polyclonal) | Immunogenes | | 1:1000 immunofluorescence |
| Antibody | Anti-CB1R (rabbit polyclonal) | Frontier science | CB1-Rb-Af380-1 | 1:500 western blot |
| Antibody | Anti-NeuN (mouse monoclonal) | Merck Millipore | MAB377 | 1:1000 |
| Antibody | Anti-rabbit IgG (H+L)-AlexaFluor-555 (donkey polyclonal) | Thermo Fisher Scientific | A-31572 | 1:1000 |
| Antibody | Anti-mouse IgG (H+L)-AlexaFluor-488 (goat polyclonal) | Jackson ImmunoResearch | 115-545-003 | 1:1000 |
| Commercial assay, kit | Immobilon Forte Western HRP Substrate | Merck Millipore | WBLUF0500 | |
| Commercial assay, kit | Nucleospin RNA isolation kit | Macherey-Nagel | 740955-250 | |
| Commercial assay, kit | SuperScript III enzyme | Invitrogen | 12574-026 | |
| Commercial assay, kit | SybrGreen master mix | Thermo Fisher | 4309155 | |
| Sequence-based reagent | Primers *Cnr1* (CB1R) (forward) | Sigma-Aldrich | | 5'-CCTGGGAAGTGTCATCTTTGT-3' |
| Sequence-based reagent | Primers *Cnr1* (CB1R) (reverse) | Sigma-Aldrich | | 5'-GGTAACCCCACCCAGTTTGA-3' |

*Continued on next page*

*Continued*

| Reagent type (species) or resource | Designation | Source or reference | Identifiers | Additional information |
|---|---|---|---|---|
| Sequence-based reagent | Primers *Gapdh* (GAPDH) (forward) | Sigma-Aldrich | | 5'-TGTCGTGGAGTCTACTGGTGTCTT-3' |
| Sequence-based reagent | Primers *Gapdh* (GAPDH) (reverse) | Sigma-Aldrich | | 5'-TGGCTCCACCCTTCAAGTG-3' |
| Sequence-based reagent | Primers *Hprt1 (HPRT)* (forward) | Sigma-Aldrich | | 5'-AAGCTTGCTGGTGAAAAGGA-3' |
| Sequence-based reagent | Primers *Hprt1 (HPRT)* (reverse) | Sigma-Aldrich | | 5'-TTGCGCTCATCTTAGGCTTT-3' |
| Chemical compound, drug | DAPI Fluoromount-G mounting media | Thermo Fisher Scientific | 00-4959-52 | |
| Software, algorithm | GraphPad Prism 7 | GraphPad Software, Inc | RRID:SCR_002798 | |
| Software, algorithm | STATISTICA 6.0 | StatSoft, USA | RRID:SCR_014213 | |
| Software, algorithm | Smart 3.0 videotracking software | Panlab | RRID:SCR_002852 | |
| Software, algorithm | ImageJ | National Institutes of Health, Bethesda, Maryland, USA | RRID:SCR_003070 | |
| Software, algorithm | The Quantity One software v4.6.3 | Bio-Rad | RRID:SCR_014280 | |
| Software, algorithm | Salmon | PMID:28263959 | RRID:SCR_017036 | v0.7.2 |
| Software, algorithm | R | https://www.R-project.org/ | RRID:SCR_001905 | v3.6.3 |
| Software, algorithm | tximport | PMID:26925227 | RRID:SCR_016752 | v1.2.0 |
| Software, algorithm | DESeq2 | PMID:25516281 | RRID:SCR_015687 | v1.26.0 |
| Software, algorithm | clusterProfiler | PMID:22455463 | RRID:SCR_016884 | v3.14.3 |
| Chemical compound, drug | SR141716A | Tocris | 158681-13-1 | |
| Chemical compound, drug | SR144528 | Tocris | 192703-06-3 | |
| Chemical compound, drug | CP55,940 | Tocris | 83002-04-4 | |
| Chemical compound, drug | WIN55,212–2 | Sigma-Aldrich | 131543-23-2 | |
| Chemical compound, drug | Tris–HCl | Sigma-Aldrich | 1185-53-1 | |
| Chemical compound, drug | BSA | Sigma-Aldrich | 9048-46-8 | |
| Chemical compound, drug | HEPES | Sigma-Aldrich | 7365-45-9 | |
| Chemical compound, drug | NaCl | Sigma-Aldrich | 7647-14-5 | |
| Chemical compound, drug | $MgCl_2$ | Sigma-Aldrich | 7791-18-6 | |
| Chemical compound, drug | EGTA | Sigma-Aldrich | 13368-13-3 | |
| Chemical compound, drug | GDP | Sigma-Aldrich | 43139-22-6 | |

*Continued on next page*

*Continued*

| Reagent type (species) or resource | Designation | Source or reference | Identifiers | Additional information |
|---|---|---|---|---|
| Chemical compound, drug | DTT | Sigma-Aldrich | 3483-12-3 | |
| Chemical compound, drug | GTPγS | Sigma-Aldrich | 10220647001 | |
| Chemical compound, drug | [³H]CP55,940 | PerkinElmer | NET1051250UC | |
| Chemical compound, drug | [³⁵S]GTPγS | PerkinElmer | NEG030H250UC | |
| Other | β-Radiation sensitive film | Sigma-Aldrich | F5263-50EA | See autoradiography methodology |

## Animals

CD mice were obtained as previously described (*Segura-Puimedon et al., 2014*) and maintained on C57BL/6J background (backcrossed for nine generations). WT littermates were used as controls. Male mice aged between 8 and 16 weeks were used for experiments and groups were balanced by age. In order to test social behavior juvenile (4 weeks old) male C57BL/6J mice were used as stranger mice.

Mice were housed in controlled environmental conditions (21 ± 1°C temperature and 55 ± 10% humidity) and food and water were available ad libitum. All the experiments were performed during the light phase of a 12 hr light/dark cycle (light on at 8 AM; light off at 8 PM). Mice were habituated to the experimental room and handled for 1 week before starting the experiments. All behavioral experiments were conducted by an observer blind to the experimental conditions.

## Drug treatment

JZL184 (Abcam) was diluted in 15% dimethyl sulfoxide (Scharlau Chemie), 4.25% polyethylene glycol 400 (AppliChem), 4.25% Tween-80 (Sigma-Aldrich), and 76.5% saline. JZL184 was injected in a volume of 5 ml/kg of body weight, respectively. Drugs were administered daily by i.p. injection 2 hr prior behavioral testing.

## Behavioral tests

### Sociability and preference for social novelty

Social behavior was performed in the V-social-maze (30 cm long × 4.5 cm wide × 15 cm height each corridor) as previously described (*Martínez-Torres et al., 2019*). Briefly, the protocol consists of three phases: habituation (Phase I), sociability (Phase II), and preference for social novelty (Phase III). First, experimental mice were introduced into the central part of the V-maze where they freely explored the two empty chambers at the end of the corridors. This measurement is important to discard a possible bias for one particular chamber and it informs about the baseline activity of the mouse in the maze. Then, during the sociability phase an unfamiliar juvenile mouse assigned as stranger 1, was placed in one of the chambers (both sides were alternated during the experiments). The experimental mouse was allowed to explore both compartments for 5 min. The experimenter recorded the time that the experimental mouse spent exploring the empty chamber or the stranger 1. Finally, the preference for social novelty phase was performed just after the sociability session. A second novel juvenile mouse, assigned as stranger 2, was placed inside the previously empty chamber, while the stranger 1 remained inside the same chamber as in Phase II. For 5 min, the experimental animal was allowed to explore the two strangers and the time spent exploring each stranger was recorded.

The procedure was performed in a sound-attenuated room with dim illumination 5–10 lux. A digital camera on top of the maze was used to record the sessions. Social exploration was considered when the experimental mouse directed the nose in close proximity (1 cm) to the vertical bars of the chambers. Mice that explored <10 s both mice were excluded from the analysis.

Acute treatment and last administration of the subchronic treatment (10 days) of JZL184 were performed 2 hr before the V-social-maze test.

In order to assess the exploratory behavior toward objects in the same setting, the same procedure was performed using objects instead of juvenile stranger mice.

## Locomotor activity

After 9 days of treatment with vehicle or JZL184, locomotor activity was assessed 2 hr after the last administration. Spontaneous locomotor responses were assessed for 30 min by using individual locomotor activity boxes (9 cm width × 20 cm length × 11 cm high, Imetronic) in a low luminosity environment (5 lux). The total activity (number of horizontal movements) was detected by a line of photocells located 2 cm above the floor.

## Novel object-recognition test

The NORT was performed as described before (*Puighermanal et al., 2009*) in a V-shaped maze (V-maze). On day 1, mice were habituated to the empty V-maze for 9 min (habituation phase). On day 2, two identical objects (familiar objects) were presented at the end of each corridor of the V-maze and mice were left to explore for 9 min before they were returned to their home cage (familiarization phase). After 10 min for short-term memory, or 24 hr for long-term memory, mice were placed back in the V-maze where one of the familiar objects was replaced by a new object (novel object) in order to assess memory performance (test phase). The time spent exploring each of the objects (familiar and novel) during the test session was computed to calculate a discrimination index (DI = (TIMEnovel − TIMEfamiliar)/(TIMEnovel + TIMEfamiliar)), defining exploration as the orientation of the nose toward the object within 2 cm from the object and with their nose facing it. A higher discrimination index is considered to reflect greater memory retention for the familiar object. Drug administration was performed 2 hr before the habituation and the training phases the 6th and 7th day of the subchronic treatment, respectively.

## Elevated plus maze

The elevated plus maze was performed to measure anxiety-like behavior as previously described (*Escudero-Lara et al., 2020*). The test consisted of a black Plexiglas apparatus with four arms (29 cm long × 5 cm wide), two closed arms with walls (20 cm high) and two open arms, set in cross from a neutral central square (5 × 5 cm) elevated 40 cm above the floor. Light intensity in the open and closed arms was 45 and 5 luxes, respectively. Mice were placed in the central square facing one of the open arms and tested for 5 min. The percentage of time spent in the open arms was determined as a measure of anxiety. The total entries in both arms were measured as a control for locomotion. Animals that exit the maze during exploration were excluded. Drug administration was performed 2 hr before the task on the 10th day of the subchronic treatment.

## Context fear conditioning

Context recognition memory was assayed in a conditioning chamber with an electrifiable floor, as previously described (*Gomis-González et al., 2021*). On training phase, mice were placed in the shuttle box, and after a period of free exploration, mice received a footshock (unconditioned stimulus [US]: 2 s, 0.35-mA intensity). Freezing behavior (lack of movement except for respiration) due to context reexposure was assessed in the same conditioning chamber 24 hr after the conditioning session. For testing, mice were placed again in the conditioning chamber for 5 min in the absence of the shock and the freezing behavior was manually counted.

## Barnes maze

Spatial learning and reference memory were assessed using the Barnes maze, as previously described (*Gomis-González et al., 2021*). The maze consists of a circular platform (90 cm in diameter) with 20 equally spaced holes through which mice may escape from a bright light (300 lx). Only one hole allows the escape to a dark/target box. Visual cues were placed surrounding the maze for navigational reference. Smart v3.0 software was used to control the video-tracking system. Briefly, mice were first habituated to the maze. In this phase, animals were placed in the center of the maze covered by an opaque cylinder for 10 s. After removal of the opaque cylinder, mice were gently guided to the target hole by surrounding them within a cylinder with transparent walls so mice could see where the scape hole was located. Then, they were left inside the target box for 2 min and then taken to the home cage.

One hour later, the first training phase was carried out. During training, each mouse performed two trials per day on 4 consecutive days. Each training trial started with the mouse placed in the center of the Barnes maze covered by an opaque cylinder for 10 s. Then, animals were allowed to explore the maze for 3 min. During this period, the primary latency to find the target hole was measured. Each training trial ended when the mouse entered the target box or after 3 min of exploration. The mouse was allowed to stay in the target box for 1 min. When mice did not reach the target box within 3 min, the experimenter guided the mouse gently to the escape box using a transparent cylinder. On day 5, the test trial was conducted 24 hr after the last training day. During the test trial, the target hole was closed. Animals were placed in the center of the maze covered by an opaque cylinder for 10 s. Next, exploration was analyzed during 90 s to reveal the number of pokes in each hole. Using the tracking system, the time spent in each quadrant (target, opposite, left, and right quadrants) were measured.

## Endocannabinoid quantification by liquid chromatography–tandem mass spectrometry

The following *N*-acylethanolamines and 2-monoacylglycerols were quantified: AEA, DEA, DHEA, 2-AG, 2-LG, and 2-OG. Its quantification was performed as described in **Navarro-Romero et al., 2019**. Briefly, half whole brains (231.4 ± 20.38 mg) (mean ± standard deviation [SD]) of mice were homogenized with 700 µl of 50 mM Tris–HCl buffer (pH 7.4):methanol (1:1) containing 25 µl of a mix of deuterated internal standards (5 ng/ml AEA-d4, 5 ng/ml DHEA-d4, 5 µg/ml 2-AG-d5, and 10 µg/ml 2-OG-d5) dissolved in acetonitrile. Afterwards, 5 ml of chloroform was added, and samples were shacked for 20 min and centrifuged at 1700 × *g* over 5 min at room temperature. Lower organic phase was evaporated under a stream of nitrogen, reconstituted in 100 µl of a mixture water:acetonitrile (10:90, vol/vol) with 0.1% formic acid (vol/vol) and transferred to microvials for liquid chromatography analysis.

An Agilent 6410 triple quadrupole Liquid-Chromatograph equipped with a 1200 series binary pump, a column oven and a cooled autosampler (4°C) were used to separate endocannabinoids. Chromatographic separation was carried out with a Waters C18-CSH column (3.1 × 100 mm, 1.8 µm particle size). The composition of the mobile phase was: A: 0.1% (vol/vol) formic acid in water; B: 0.1% (vol/vol) formic acid in acetonitrile. Gradient chromatography was used to separate endocannabinoids and related compounds and the ion source was operated in the positive electrospray mode. The selective reaction monitoring mode was used for the analysis. Quantification was done by isotope dilution with the response of the deuterated internal standards.

## Cannabinoid receptor autoradiography

Five fresh consecutive sections from brain of CD and WT mice were dried and submerged in 50 mM Tris–HCl buffer containing 1% of bovine serum albumin (BSA) (pH 7.4) for 30 min at room temperature, followed by incubation in the same buffer in the presence of the CB1R/CB2R radioligand, [3H]CP55,940 (3 nM) for 2 hr at 37°C. Nonspecific binding was measured by competition with nonlabeled CP55,940 (10 µM) in another consecutive slice. The CB1R antagonist, SR141716A (1 µM) and the CB2R antagonist, SR144528 (1 µM), were used together with [3H]CP55,940 in two consecutive slices to check the CB1R- or CB2R-binding specificity. Then, sections were washed in ice-cold (4°C) 50 mM Tris–HCl buffer supplemented with 1% BSA (pH 7.4) to stop the binding, followed by dipping in distilled ice-cold water and drying (4°C). Autoradiograms were generated by exposure of the tissues for 21 days at 4°C to β-radiation sensitive film together with [3H]-microscales used to calibrate the optical densities to fmol/mg tissue equivalent (fmol/mg t.e.).

## Labeling of activated $G_{i/o}$ proteins by [35S]GTPγS-binding assay

Brain samples from WT and CD mice were fresh frozen, cut into 20 µm sections, mounted onto gelatin-coated slides and stored (−25 °C) until used. Six fresh consecutive sections from each mouse were dried, followed by two consecutive incubations in HEPES-based buffer (50 mM HEPES, 100 mM NaCl, 3 mM MgCl₂, 0.2 mM EGTA, and 1% BSA, pH 7.4) for 30 min at 30°C. Briefly, sections were incubated for 2 hr at 30°C in the same buffer supplemented with 2 mM GDP, 1 mM DTT, and 0.04 nM [35S]GTPγS (PerkinElmer). The [35S]GTPγS basal binding was determined in two consecutive sections in the absence of agonist. The agonist-stimulated binding was determined in a consecutive brain section in the same reaction buffer in the presence of the CB1R/CB2R agonist, WIN55,212–2 (10 µM).

The CB1R antagonist, SR141716A (1 µM) and the CB2R antagonist, SR144528 (1 µM), were used together with the agonist in two consecutive slices to check the specificity of the CB1R or CB2R functionality. Nonspecific binding was defined by competition with cold GTPγS (10 µM) in another section. Then, sections were washed twice in cold (4°C) 50 mM HEPES buffer (pH 7.4), dried, and exposed to β-radiation sensitive film with a set of [$^{14}$C] standards (American Radiolabelled Chemicals). After 48 hr, the films were developed, scanned, and quantified by transforming optical densities into nCi/g tissue equivalence units using a calibration curve defined by the known values of the [$^{14}$C] standards (ImageJ). Nonspecific binding values were subtracted from both agonist- and basal-stimulated conditions. The percentages of agonist-evoked stimulation were calculated from both the net basal and net agonist-stimulated [$^{35}$S]GTPγS-binding densities according to the following formula: ([$^{35}$S]GTPγS agonist-stimulated binding × 100/ [$^{35}$S]GTPγS basal binding) – 100.

## Tissue preparation for immunofluorescence

Immediately after social behavior assessment, a group of mice were deeply anesthetized by intraperitoneal injection of ketamine (100 mg/kg) and xylazine (20 mg/kg) mixture in a volume of 0.2 ml/10 g of body weight. Subsequently, mice were intracardially perfused with 4% paraformaldehyde in 0.1 M $Na_2HPO_4$/0.1 M $NaH_2PO_4$ buffer (PB), pH 7.5, delivered with a peristaltic pump at 19 ml/min flow for 3 min.

Afterwards, brains were extracted and postfixed in the same solution for 24 hr and transferred to a solution of 30% sucrose in PB overnight at 4°C. Coronal frozen sections (30 µm) of the basolateral amygdala (from Bregma: −1.22 to −1.82 mm) were obtained on a freezing microtome and stored in a solution of 5% sucrose at 4°C until used.

## Brain immunofluorescence and image analysis

Free-floating brain slices were rinsed in PB 0.1 M three times during 5 min with PB. Subsequently, brain slices were blocked in a solution containing 3% donkey serum (DS) (D9663, Sigma-Aldrich) and 0.3% Triton X-100 (T) in PB (DS-TPB) at room temperature for 2 hr, and incubated overnight in the same solution with the primary antibody to CB1R (1:1000, rabbit, Immunogenes) and neuronal nuclei (NeuN) (1:1000, mouse, MAB377, Merck Millipore), at 4°C. The next day, after three rinses in PB of 10 min each, sections were incubated at room temperature with the secondary antibody AlexaFluor-555 donkey anti-rabbit (for CB1R) (1:1000, A-31572, Life Technologies, Thermo Fisher Scientific) and secondary antibody AlexaFluor-488 goat anti-mouse (for NeuN) (1:1000, 115-545-003, Jackson ImmunoResearch) in DS-T-PB for 2 hr. After incubation, sections were rinsed three times for 10 min each and mounted immediately after onto glass slides coated with gelatin in Fluoromont-G with DAPI (00-4959-52, Invitrogen, Thermo Fisher Scientific) as counterstaining.

Immunostained brain sections were analyzed with a ×10 objective using a Leica DMR microscope (Leica Microsystems) equipped with a digital camera Leica DFC 300 FX (Leica Microsystems). The delimitation of basolateral amygdala area in each image was manually determined for quantification. The images were processed using the ImageJ analysis software. The mean intensity of the determined region was quantified using the automatic 'measure' option of ImageJ. Two to four representative images for each animal were quantified, and the average intensity of CB1R and the density of Dapi and NeuN + cells (cells/mm$^2$) was calculated for each mouse. The data are expressed as a percentage of the control group (WT VEH). The displayed images for CB1R were flipped for orientation consistency, adjusted for brightness and contrast and transformed to gray scale for display.

## Protein sample preparation

Immediately after social behavior assessment, amygdala and cardiac tissues were dissected from another group of mice. Tissues were frozen on dry ice and stored at −80°C until used, as previously reported. Samples from all animal groups, in each experiment, were processed in parallel to minimize interassay variations. The preparation of amygdala samples for total solubilized fraction was performed as previously described (*Gomis-González et al., 2021*). Frozen brain areas were dounce-homogenized in 30 volumes of lysis buffer (50 mM Tris–HCl pH 7.4, 150 mM NaCl, 10% glycerol, 1 mM EDTA, 10 µg/ml fluoride, 5 mM sodium pyrophosphate, and 40 mM beta-glycerolphosphate) plus 1% Triton X-100. After 10 min incubation at 4°C, samples were centrifuged at 16,000 × *g* for 20 min to

remove insoluble debris. Protein contents in the supernatants were determined by DC-micro plate assay (Bio-Rad, Madrid, Spain), following the manufacturer's instructions.

## Immunoblot analysis

Anti-CB1R (1:500, rabbit, CB1-Rb-Af380-1, Frontier science) were detected using horseradish peroxidase-conjugated anti-rabbit antibody (1:15,000, Cell Signaling Technologies) and visualized by enhanced chemiluminescence detection (Immobilon Forte Western HRP substrate, Merck Millipore). Digital images were acquired on ChemiDoc XRS System (Bio-Rad) and quantified by The Quantity One software v4.6.3 (Bio-Rad). Optical density values for target proteins were normalized to Ponceau staining of the nitrocellulose membrane as loading control, and expressed as a percentage of control group (VEH-treated mice).

## Cardiac real-time qPCR

Animals were euthanized by cervical dislocation, and immediately after, the chest cavity was opened and hearts were excised. Afterwards, they were washed in phosphate-buffered saline and atria and large vessels were removed. Ventricles were cut in small pieces, snap frozen in liquid nitrogen and kept at −80°C until use.

Frozen tissue was minced in lysis buffer from the RNA isolation kit using a homogenizer (Polytron PT2500 E) and RNA was extracted in 60 µl of RNAse-free water with the Nucleospin kit (Macherey-Nagel, #740955-250). Following extraction, 50 µg of RNA were retrotranscribed into cDNA with the SuperScript III enzyme (Invitrogen, #12574-026). CB1 gene expression was measured in a real-time quantitative PCR machine (QuantStudio 12K Flex) using the SybrGreen master mix (Thermo Fisher, #4309155) and the following predesigned gene-specific primers *Cnr1* (CB1R) (forward: CCTGGGAAGTGTCATCTTTGT, reverse: GGTAACCCCACCCAGTTTGA), *Gapdh* (GAPDH) (forward: TGTCGTGGAGTCTACTGGTGTCTT, reverse: TGGCTCCACCCTTCAAGTG), and *Hprt1* (HPRT) (forward: AAGCTTGCTGGTGAAAAGGA, reverse: TTGCGCTCATCTTAGGCTTT) were the endogenous controls used for normalization. Each sample was determined in triplicate using 5 ng of cDNA per run.

## Echocardiogram

Echocardiograms were performed 2 hr after the last administration of JZL184 subchronic treatment (10 days). Studies were carried out under general anaesthesia with isoflurane (2%) using a Vivid IQ and a rodent-specific L8-18i-D Linear Array 5–15 MHz probe (General Electric Healthcare, Horten, Norway). Mice were placed in supine position on a continuously warmed platform to maintain body temperature, and all four limbs were fixed. Ultrasound gel was applied on the left hemithorax and hearts were imaged in parasternal short-axis projections. M-mode echocardiograms of the midventricle were recorded at the level of the papillary muscles. The left ventricular end-diastolic (LVDD) and end-systolic (LVSD) internal diameters were measured in the M-mode recordings and were computed by the Teichholz formula into volumes as follows: $LVDV = 7*LVDD^3/2.4$ (where LVDV means LV end-diastolic volume) and $LVSV = 7*LVSD^3/2.4 + LVSD$ (where LVSV means LV end-systolic volume). LVEF was subsequently calculated as follows: $LVEF = ((LVDV − LVSV)/LVDV) \times 100$ and used as surrogate for LV systolic function as proposed by the American Society of Echocardiography (*Mitchell et al., 2019*). The average of three consecutive cardiac cycles was used for each measurement, with the operator being blinded to the group assignment.

## Heart histology and image analysis

Immediately after echocardiograms, mice were intracardially perfused as described before. Afterwards, hearts were paraffin embedded and serial transverse sections (8 µm) were collected on a glass slide. Sections were stained with a regular hematoxylin–eosin (H&E) protocol. Images of H&E-stained heart samples were obtained using visible light with an Olympus DP71 camera attached to an Olympus MVX10 MacroView Upright Microscope (zoom factor 1.25). The percentage of muscle was calculated by subtracting the lumen area from the total area occupied by the heart and indexing to the total area. Measures were performed with ImageJ software.

## Cardiomyocyte analysis

To evaluate cellular hypertrophy, random photomicrographs of each H&E-stained heart section were taken using a Leica DM 6000B microscope coupled with a Hamamatsu ORCA-ER C4742-80 camera. Left ventricle CSA was measured by delimitating the outer diameter of the cardiomyocytes' transverse section. At least 20–30 cells per slide were measured using the ImageJ software. All images were taken and measured blinded.

## Blood pressure assessment

Systolic, mean, and diastolic blood pressure were measured in conscious mice on three separate occasions by using a tail cuff system (Non-Invasive Blood Pressure System, PanLab), while holding the mice in a black box on a heated stage. In order to improve measurement consistency, multiple sessions were performed to train each mouse. At least 9 readings (3 per session) were made for each mouse.

## RNA-seq analysis

Total RNA from heart was extracted following the standard protocol using TRIZOL reagent (Invitrogen) according to the manufacturer's instructions. RNA-seq libraries were generated for Illumina sequencing with a HiSeq3000 instrument. The RNA-seq datasets were deposited on the Gene Expression Omnibus (GEO) and made publicly available through the accession GSE164257.

Transcript abundances in Transcripts Per Million (TPM) were quantified with Salmon 0.7.2 using the Ensembl annotation GRCm38 v85. Read counts for genes were obtained from Salmon quantification with tximport (v 1.2.0) R package (*Soneson et al., 2015*) and genes with less than 1 logCPM were filtered out. For the differential expression analysis, the count matrix was used with DESeq2 (v 1.26.0) Bioconductor package (*Love et al., 2014*) as input directly from the tximport package. DESeq2 was used to estimate fold-changes and p values for each gene between conditions and p values were corrected using the Wald test procedure. The DEGs were selected based on a $|\log_2$ fold-change$| > 0$ and adjusted p value <0.05. Functional analysis was performed to examine the biological processes of the DEGs with clusterProfiler (v 3.14.3) Bioconductor package (*Yu et al., 2012*) Benjamini–Hochberg test was used to adjust the enrichment p value for the GO terms in each defined gene set using the expressed genes as background. Enriched GO terms were determined based on adjusted p value <0.05. Cardiovascular genes were selected from those genes with a described GO biological process term related with cardiovascular function.

For cardiac-related process analysis, we first used all genes in WT and CD treated that were statistically significant differentially expressed ($p < 0.05$ and $|\log2FC| > 0$). We searched in REACTOME database for pathways directly implicated in cardiac processes and crossed the results with this DEG list. Then, for those genes from CD and WT mice that appear in the selected pathways, we look if there is a change in their fold-change in the comparison of CD mice treated for 10 days with JZL184 (8 mg/kg) or vehicle. Only the fold-change of those genes that were significant differentially expressed were shown.

## Statistics

Mice were randomly assigned to experimental groups. Sample size choice was based on previous studies (*Navarro-Romero et al., 2019*; *Busquets-Garcia et al., 2013*) and it is listed in figure legends for each experiment. Data were analyzed with Statistica Software and GraphPad Prism 7 using unpaired Student's *t*-test and two-way analysis of variance (ANOVA) for multiple comparisons. Social interaction was analyzed by repeated measures ANOVA with maze/genotype/treatment as between-subject factor and compartment as within-subject factor. Subsequent post hoc analysis (Newman–Keuls) was used when required (significant effect of factors or interaction between factors). Comparisons were considered statistically significant when $p < 0.05$. Outliers (±2 SD from the mean) were excluded. The artwork was designed using GraphPad Prism 7. All results were expressed as mean ± standard error of the mean.

## Ethics

All animal procedures were conducted following ARRIVE (Animals in Research: Reporting In Vivo Experiments) guidelines (*Kilkenny et al., 2010*) and standard ethical guidelines (European Union Directive 2010/63/EU) and approved by the local ethical committee (Comitè Ètic d'Experimentació Animal-Parc de Recerca Biomèdica de Barcelona, CEEA-PRBB).

## Acknowledgements

We are grateful to Marta Linares, Dulce Real, Raquel Martín, Jolita Jancyte, and Francisco Porrón for expert technical assistance and the Laboratori de Neurofarmacologia-NeuroPhar for helpful discussion. A.N.-R. was the recipient of a predoctoral fellowship (Ministerio de Educación y Cultura, Spain), L.G.-L. was supported by predoctoral fellowship by FPI (MINE-ICO/FEDER, EU). This study was supported by Ministerio de Economía, Innovación y Competitividad (MINECO), Spain #RTI2018-099282-B-I00B to A.O., #SAF2017-84060-R to R.M.; Generalitat de Catalunya, Spain (2017SGR-669 to R.M.); Ministerio de Ciencia e Innovación (SAF2016-78508-R; AEI/MINEICO/FEDER, UE) to VC. Basque Government IT1454-22 to the 'Neurochemistry and Neurodegeneration' consolidated research group to R R-P. Instituto de Salud Carlos III (PI20/00153, co-funded by the European Union [ERDF 'A way to make Europe']) to R R-P. ICREA (Institució Catalana de Recerca i Estudis Avançats, Spain) Academia to A.O. and R.M. Grant 'Unidad de Excelencia María de Maeztu', funded by the MINECO (#MDM-2014-0370); IPEP MdM 2017 to A.O. and E.E. FEDER, European Commission funding is also acknowledged.

## Additional information

### Competing interests

Eduardo Eyras: Reviewing editor, *eLife*. The other authors declare that no competing interests exist.

### Funding

| Funder | Grant reference number | Author |
|--------|------------------------|--------|
| Ministerio de Ciencia, Innovación y Universidades | FPU13/01867 | Alba Navarro-Romero |
| Ministerio de Ciencia e Innovación | BES-2016-077950 | Lorena Galera-López |
| Ministerio de Ciencia e Innovación | RTI2018-099282-B-I00 | Andres Ozaita |
| Ministerio de Ciencia e Innovación | SAF2017-84060-R | Rafael Maldonado |
| Generalitat de Catalunya | 2017SGR-669 | Rafael Maldonado |
| Ministerio de Ciencia e Innovación | SAF2016-78508-R | Victoria Campuzano |
| Basque Country Government | IT1454-22 | Rafael Rodríguez-Puertas |
| Instituto de Salud Carlos III | PI20/00153, co-funded by the European Union (ERDF "A way to make Europe") | Rafael Rodríguez-Puertas |
| Institució Catalana de Recerca i Estudis Avançats | | Rafael Maldonado |
| Ministerio de Economía y Competitividad | #MDM-2014-0370 | Alba Navarro-Romero |
| Ministerio de Economía y Competitividad | #MDM-2014-0370 | Eduardo Eyras |
| Ministerio de Ciencia e Innovación | PRE2019-087644 | Lucía de los Reyes-Ramírez |

| Funder | Grant reference number | Author |
| --- | --- | --- |
| Ministerio de Ciencia e Innovación | AEI/MINEICO/FEDER | Andres Ozaita |
| Ministerio de Ciencia e Innovación | AEI/MINEICO/FEDER | Rafael Maldonado |
| Ministerio de Ciencia e Innovación | AEI/MINEICO/FEDER | Victoria Campuzano |
| Ministerio de Economía y Competitividad | FEDER | Alba Navarro-Romero |
| Ministerio de Economía y Competitividad | FEDER | Eduardo Eyras |
| Ministerio de Economía y Competitividad | IPEP MdM-2017 | Eduardo Eyras |
| Ministerio de Ciencia e Innovación | UE | Andres Ozaita |
| Ministerio de Ciencia e Innovación | UE | Rafael Maldonado |
| Ministerio de Economía y Competitividad | UE | Alba Navarro-Romero |
| Ministerio de Ciencia e Innovación | UE | Victoria Campuzano |
| Ministerio de Economía y Competitividad | EU | Eduardo Eyras |
| FRAXA Research Foundation | | Lorena Galera-López |

The funders had no role in study design, data collection, and interpretation, or the decision to submit the work for publication.

## Author contributions

Alba Navarro-Romero, Data curation, Formal analysis, Investigation, Methodology, Writing – original draft, Writing – review and editing, participated in experimental design; Lorena Galera-López, Data curation, Formal analysis, Investigation, Methodology, Writing – review and editing, participated in experimental design; Paula Ortiz-Romero, Investigation, Methodology, Writing – review and editing, conducted and analyzed histological and molecular experiments; Alberto Llorente-Ovejero, Data curation, Formal analysis, Investigation, Methodology, Writing – review and editing, conducted and analyzed autoradiography experiments; Lucía de los Reyes-Ramírez, Data curation, Formal analysis, Investigation, Writing – review and editing, analyzed and interpreted transcriptomic data and posted transcriptomic data in the public repository; Iker Bengoetxea de Tena, Formal analysis, Investigation, conducted and analyzed autoradiography experiments; Anna Garcia-Elias, Formal analysis, Investigation, conducted histological analysis and q-RT-PCR analysis; Aleksandra Mas-Stachurska, Data curation, Investigation, Methodology, Writing – review and editing, conducted echocardiography analysis; Marina Reixachs-Solé, Data curation, Formal analysis, Investigation, Methodology, Writing – review and editing, analyzed and interpreted transcriptomic data; Antoni Pastor, Formal analysis, Investigation, Methodology, Writing – review and editing, performed endocannabinoid level measures; Rafael de la Torre, Resources, Methodology, Writing – review and editing, performed endocannabinoid level measures; Rafael Maldonado, Resources, Methodology, Writing – review and editing, participated in the supervision and experimental design; Begoña Benito, Resources, Data curation, Supervision, Methodology, Writing – review and editing, participated in electrocardiogram measures and histological interpretation; Eduardo Eyras, Conceptualization, Resources, Supervision, Funding acquisition, Visualization, Methodology, Writing – review and editing, analyzed the transcriptomic data and provided input into their interpretation; Rafael Rodríguez-Puertas, Resources, Methodology, Writing – review and editing, performed and analyzed autoradiography experiments; Victoria Campuzano, Resources, Formal analysis, Funding acquisition, Methodology, Writing – review and editing, provided complete deletion (CD) mouse line; Andres Ozaita, Conceptualization, Resources,

Supervision, Funding acquisition, Investigation, Methodology, Writing – original draft, Project administration, Writing – review and editing, conceptualized

### Author ORCIDs
Aleksandra Mas-Stachurska http://orcid.org/0000-0003-4947-4111
Rafael Maldonado http://orcid.org/0000-0002-4359-8773
Andres Ozaita http://orcid.org/0000-0002-2239-7403

### Ethics

This study was conducted following ARRIVE (Animals in Research: Reporting In Vivo Experiments) and standard ethical guidelines (European Union Directive 2010/63/EU) and procedures approved by the local ethical committee (Comitè Ètic d'Experimentació; Animal-Parc de Recerca Biomèdica de Barcelona, CEEA-PRBB) and local authorities (Generalitat de Catalunya, Dept d'Acció; Climàtica, Alimentació i Agenda Rural). Every effort was made to minimize animal suffering.

### Decision letter and Author response
Decision letter https://doi.org/10.7554/eLife.72560.sa1
Author response https://doi.org/10.7554/eLife.72560.sa2

## Additional files

### Supplementary files

• Supplementary file 1. [$^3$H]CP55,940 binding of all analyzed brain regions (WT, $n = 11$; complete deletion [CD], $n = 10$) in fmol/mg t.e. of cannabinoid type-1 receptor (CB1R). Statistical significance was calculated by Student's $t$-test. *$p < 0.05$; **$p < 0.01$ (genotype effect). Data are expressed as mean ± standard error of the mean (SEM).

• Supplementary file 2. [$^{35}$S]GTPγS binding evoked by WIN55,212–2 (10 µM) of all analyzed brain regions (WT, $n = 11$; complete deletion [CD], $n = 10$), expressed as percentage of stimulation over the basal binding. Statistical significance was calculated by Student's $t$-test. *$p < 0.05$; **$p < 0.01$ (genotype effect). Data are expressed as mean ± standard error of the mean (SEM).

• Supplementary file 3. Additional echocardiogram measurements after JZL184 treatment in complete deletion (CD) mice. Interventricular septum diastolic (IVSd), left ventricular posterior wall thickness diastolic (LVPWd), LV end-diastolic diameter (LVDd), and left ventricular mass diastolic (Lvmass) measurement relatives to body weight (BW) (WT VEH, $n = 8$; WT JZL184, $n = 7$; CD VEH, $n = 7$; CD JZL184, $n = 7$). Statistical significance was calculated by Newman–Keuls post hoc test following two-way analysis of variance (ANOVA) ***$p < 0.001$ (genotype effect). Data are expressed as mean ± standard error of the mean (SEM).

• Supplementary file 4. Summary of gene expression reversion after JZL 184 administration in complete deletion (CD) mice of genes implicated in cardiac-related process. Pathways were obtained from Reactome database and expression changes are expressed as $\log_2$ ratios of fold-changes between CD mice treated for 10 days with JZL184 (8 mg/kg) and CD or WT littermates treated with vehicle.

• Transparent reporting form

### Data availability

Behavioral, biochemical, immunohistochemical, echocardiography, and in situ radioligand binding data generated are available as Source data. Sequencing data have been deposited in GEO under accession code GSE164257. To review GEO accession GSE164257 associated with our submission: https://www.ncbi.nlm.nih.gov/geo/query/acc.cgi?acc=GSE164257.

The following dataset was generated:

| Author(s) | Year | Dataset title | Dataset URL | Database and Identifier |
|-----------|------|---------------|-------------|-------------------------|
| Navarro-Romero A, Galera-López L, Ortiz-Romero P, Reixachs-Solé M, de los Reyes-Ramírez L, Eyras E, Campuzano V, Ozaita A | 2021 | JZL184 treatment restores neurological and cardiac phenotypes of a mouse model of Williams-Beuren syndrome | https://www.ncbi.nlm.nih.gov/geo/query/acc.cgi?acc=GSE164257 | NCBI Gene Expression Omnibus, GSE164257 |

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
