## [Editor Report]

In this report, alterations of the brain cannabinoid type-1 receptor (CB1R) were identified in a mouse model of Williams-Beuren syndrome (CD mice). Modulation of CB1R by JZL184 treatment improved social and cognitive phenotypes and also cardiac function of CD mice. These are important and novel findings highlight a role of the endocannabinoid system and its dysfunction in Williams-Beuren syndrome (WBS) that provide a basis for future developments of potential therapeutics for WBS patients. This study will be of great interest to researchers and clinicians in the field of human genetic diseases.

---

## [Decision Letter]

**Decision letter after peer review:**

Thank you for submitting your article "Cannabinoid signaling modulation through JZL184 restores key phenotypes of a mouse model for Williams-Beuren syndrome" for consideration by *eLife*. Your article has been reviewed by 3 peer reviewers, one of whom is a member of our Board of Reviewing Editors, and the evaluation has been overseen by a Senior Editor. The following individual involved in review of your submission has agreed to reveal their identity: Jung Kwang Mook (Reviewer #3).

The reviewers have discussed their reviews with one another, and the Reviewing Editor has drafted this letter to help you prepare a revised submission.

Essential revisions:

The authors of this manuscript explored the ECS properties in a mouse model for WS. This is a novel research field in Williams-Beuren syndrome, which is of high interest and relevance. The authors improved behavioral aspects by modulating the ECS, and also demonstrate improvement of cardiovascular function. Advancing our understanding of the ECS in Williams-Beuren syndrome is important given the fact that the treatment options for Williams-Beuren syndrome are still limited. Thus, this manuscript has potential to be ground-breaking. The use of the MAGL inhibitor, based on the ECS characterization and the alterations found in CD mice compared to control animals, is elegant and well-designed. The mechanisms how the treatment affects behavior and cardiovascular function are still unclear.

1. In general, the reviewer has difficulties with the test mice age range, being 8-16 weeks of age. This is relatively a wide range, that can lead to the unclear behavioral and other phenotypes described in the manuscript. Many studies showed dynamic changes in the developmental trajectory of the endocannabinoid system in the mouse brain. It might be that the 8-16 weeks of age range is responsible for the different stages of the endocannabinoid system maturation. This is a major issue in this study in the reviewer's opinion.

2. In the novel object test, why did the authors decided to exclude from the analysis mice that explored <10s both objects? How many mice were excluded based on this criteria? Raw data should be included to allow the reader better judgement of the raw data (with all excluded mice included) and the presented final graphs.

3. In the novel object test, Figure 1c,d make no sense to me: the CD mice should have higher (!) discrimination index than WT based on Figure 1c, right, while Figure 1d shows the opposite.

4. The behavioral characterization in this study is very limited. Why the authors did not characterize other aspects related to WS, such as anxiety and motor capabilities? Tests such as elevated zero maze, open field exploration, rotarod, etc. will better define mice properties and the effectiveness of the drug to modulate these. Especially given the fact that previous article from this group showed motor deficits in CD mice (Segura-Puimedon et al., 2014) , Figure 5a-b. Also contextual fear conditioning can add more insights on fear and cognition in these mice with and without the drug. Similarly, short (i.e. 1 hour) and long-term memory (i.e. 24 hour) should be studied in the NOR test, to better define the results in Figure 1c,d (see comment below).

5. Can the authors show that the i.p. administration of the drug resulted in increased drug concentration in the brain of treated mice control to placebo? The reported alterations following drug administration might be secondary to other drug-related responses.

6. From Figure 3c in this manuscript it seems that the drug has negative effects on NOR discrimination index in WT animals (together with the positive effects on CD mice). This should be discussed in the manuscript to emphasize the potential negative effects of the drug.

7. Why did the authors not characterize the signaling of the CB1R following drug treatment in Figure 3 as they did in Figure 2? This should be demonstrated, to further study the drug's effect on the ECS.

8. The effect of the drug on the cardiovascular phenotype is not clear. There is no mechanistic explanation for this change. RNAseq is only partial aspect of this surprising effect, and is not sufficient to understand and support the mechanistic explanation of how the drug affected cardiovascular properties in only a few days. Why should it affect heart properties, especially given such a short treatment of only several days? This should be further demonstrated and explained, with more histological and molecular and cellular evidence to support this surprising effect. Moreover, did these changes in weight etc. were specific to the heart only? The different number of mice in each of the tests presented in Figure 4 is unjustified and makes the reviewer feel uncomfortable. Why did the author have to use 20 mice in Figure 4b, but used 7 in Figure 4c? Can they show statistical power calculation to support the huge difference in mice number between the tests?

9. In Figure 4B, the muscle proportion measurement method in this paper is uncommon. Heart state whether in systolic or diastolic condition at the time of sacrifice and also paraffin processing possibly produce artifacts that will affect the heart morphology and ventricular wall thickness. It will be better if the researcher showed cardiac hypertrophy by echocardiography parameters such as PWTd (posterior wall thickness diastolic) PWTs (posterior wall thickness systolic).

10. Since WBS cardiac phenotype includes hypertension, we suggest the researcher do blood pressure measurement in CD mice, also to confirm the phenotype from previous paper by Segura-Puimedon et al., Hum Mol Genet 2014.

11. Did the authors try acute or subchronic CB1 inverse agonists? If not, what would be the benefit of subchronic JZL184 than pharmacological CB1 inhibition? As a related discussion, do the authors expect any side (reverse) effect during the early phase of JZL184 treatment, maybe first a few days, when 2-AG signaling must have been elevated but changes in CB1 receptor expression is not occurred yet?

12. Endocannabinoid levels quantified from discrete brain regions closely related to the behavioral tests could be a good addition. Also, usually, the absolute values rather than comparative units are displayed for these kind of quantitation results (maybe shown in Table 1 is %? The unit was not clearly defined).

13. Does JZL184-induced normalization CB1 receptors density observed in hippocampus or heart that are related to other phenotypes (NORT or cardiac hypertrophy, respectively) that were normalized by JZL184? Otherwise, is it possible that the beneficial effects are actually a result from long-term elevation of 2-AG signaling? If these results are not available, such limitation in interpretation should be discussed.

---

## [Author Response]

Essential revisions:The authors of this manuscript explored the ECS properties in a mouse model for WS. This is a novel research field in Williams-Beuren syndrome, which is of high interest and relevance. The authors improved behavioral aspects by modulating the ECS, and also demonstrate improvement of cardiovascular function. Advancing our understanding of the ECS in Williams-Beuren syndrome is important given the fact that the treatment options for Williams-Beuren syndrome are still limited. Thus, this manuscript has potential to be ground-breaking. The use of the MAGL inhibitor, based on the ECS characterization and the alterations found in CD mice compared to control animals, is elegant and well-designed. The mechanisms how the treatment affects behavior and cardiovascular function are still unclear.1. In general, the reviewer has difficulties with the test mice age range, being 8-16 weeks of age. This is relatively a wide range, that can lead to the unclear behavioral and other phenotypes described in the manuscript. Many studies showed dynamic changes in the developmental trajectory of the endocannabinoid system in the mouse brain. It might be that the 8-16 weeks of age range is responsible for the different stages of the endocannabinoid system maturation. This is a major issue in this study in the reviewer's opinion.

We understand the concern of the Reviewer about the age of mice. Although globally 8-16 weeks seems a wide age range, we did all our efforts to compare in each experiment, littermate mice with similar ages, to prevent any age bias. In the new version of the manuscript, we indicate now that mice in experiments were always balanced for genotype and age (Methods, Line 419).

Age-related changes in the components of the endocannabinoid system have been analyzed in different studies using rodent models. CB1R density and function and the levels of the main endocannabinoids have been described to be increased, reduced or remain without changes in different brain regions (Berrendero et al., 1998; Ginsburg and Hensler, 2022; Liu et al., 2003; Maccarrone et al., 2002; Wang et al., 2003). These discrepancies may be due to the brain regions analyzed, the age of the animals, the models used or even the experimental procedures employed. In our experimental set, we did not find the age to be a relevant factor. We have analyzed the age of mice that were used on CB1R density, CB1R functional coupling to Gi/o proteins or endocannabinoid levels analysis (Author response image 1) and we have not observed significant differences in the age between genotypes.

**Author response image 1. sa2fig1:** Age distribution of mice used for the characterization of the endocannabinoid system.

2. In the novel object test, why did the authors decided to exclude from the analysis mice that explored <10s both objects? How many mice were excluded based on this criteria? Raw data should be included to allow the reader better judgement of the raw data (with all excluded mice included) and the presented final graphs.

We normally exclude those extremely rare mice with low (<10 sec) total exploration time during the test session of the novel object-recognition test (Gomis-González et al., 2021; Navarro-Romero et al., 2019) since it has been demonstrated that a minimum amount of exploration is required for reliable discrimination between the familiar and the novel object (Akkerman et al., 2012). However, after close examination of the data in the present manuscript, we realized that in this study there were no mice excluded because of this criterion, so we have removed the referred sentence from the methods section.

3. In the novel object test, Figure 1c,d make no sense to me: the CD mice should have higher (!) discrimination index than WT based on Figure 1c, right, while Figure 1d shows the opposite.

Figure 1c refers to an experiment specifically designed to assess motivation to explore object novelty. In this case, we took advantage of the Vsoc-maze employed for the social behavior (Figure 1a) but making use of novel objects (Object 1 and Object 2) instead of stranger mice (Stranger mouse 1 and Stranger mouse 2). The results in Figure 1c correspond to the exploration times in three consecutive phases performed in 15 min (Phase I, Phase II and Phase III, each 5 min) without any inter-phase interval. As displayed in Figure 1c, we observed that CD mice were able to discriminate between Object 1 (firstly encountered) and Object 2 (secondly encountered), and therefore in Phase II and Phase III CD and WT mice show similar preference for object novelty. In contrast, Figure 1d shows the novel object recognition test performed in the V-maze. This test consists of a habituation phase, a familiarization/training phase, and a test phase (each of 9 min), with an inter-phase interval of 10 min between the training and the test, where the animal goes back to the homecage, to assess short-term memory 10 min after the training phase (see Figure 1—figure supplement 1b for scheme). Therefore, the results in Figure 1c and 1d suggest that CD mice have preference for object novelty but when there is an interval of 10 min between training and test phases CD mice do not discriminate novelty which is interpreted as a deficit in short-term memory. To clarify this extent, we have added a new panel including a schematic cartoon of short-term object recognition memory in Figure 1—figure supplement 1b.

4. The behavioral characterization in this study is very limited. Why the authors did not characterize other aspects related to WS, such as anxiety and motor capabilities? Tests such as elevated zero maze, open field exploration, rotarod, etc. will better define mice properties and the effectiveness of the drug to modulate these. Especially given the fact that previous article from this group showed motor deficits in CD mice (Segura-Puimedon et al., 2014) , Figure 5a-b. Also contextual fear conditioning can add more insights on fear and cognition in these mice with and without the drug. Similarly, short (i.e. 1 hour) and long-term memory (i.e. 24 hour) should be studied in the NOR test, to better define the results in Figure 1c,d (see comment below).

Following the Reviewer’s comments, we have explored several other behavioral traits. We investigated the anxiety-like responses in the elevated-plus maze where we found an effect of JZL184 in the anxiety phenotype in CD mice. JZL184 repeated administration produced an anxiogenic effect in CD mice, that turned CD behavior to WT control levels. We have now included this result in the manuscript Figure 3c

We further characterized the cognitive profile of CD mice. In particular, the long-term novel object-recognition memory test to assess low-arousal object-recognition memory, the context fear conditioning to study emotional memory and the Barnes maze test to study spatial learning and memory. Interestingly, we did not observe significant differences between CD and WT mice in any of these tests, now displayed in Figure 1—figure supplement 2, so we did not further assess the effect of the treatment in these paradigms.

Regarding novel object-recognition memory, our new data on long-term memory reveals, together with the alterations already reported in short-term memory, better pinpoint the deficits in CD mice. These new data are displayed in Figure 1—figure supplement 2a. Regarding context fear-conditioning, our new data show no significant differences between genotypes, reminiscent of those previously reported using a similar approach (Segura-Puimedon et al., 2014). These new data are reported in Figure 1—figure supplement 2b. Finally, the Barnes maze test did not reveal significant differences between genotypes in spatial learning and memory, somehow resembling the results previously described for spatial memory in the Morris water maze (Segura-Puimedon et al., 2014). These new data are displayed in Figure 1—figure supplement 2, c and d.

5. Can the authors show that the i.p. administration of the drug resulted in increased drug concentration in the brain of treated mice control to placebo? The reported alterations following drug administration might be secondary to other drug-related responses.

We appreciate the Reviewer’s comment. We focused in assessing the levels of the endocannabinoid 2-AG in the main brain regions of interest (amygdala and hippocampus) after treatment. We observed that repeated i.p. administration of JZL184 significantly increases between seven times in the case of the amygdala, and nine times, in the case of the hippocampus, the levels of 2-AG, but not those of anandamide. Both CD and WT mice showed similar levels of 2-AG after JZL184 treatment. We have included this new analysis in Table 2. While moderate increases of 2-AG levels (two-fold increases) were not observed to modify CB1R activity (Schlosburg et al., 2010), higher increases on 2-AG content were found to produce CB1R downregulation and desensitization (Chanda et al., 2010; Schlosburg et al., 2010). Therefore, the downregulation/desensitization of CB1R in specific brain regions in our experimental conditions could be associated to the strong upregulations of 2-AG levels (seven-to-nine-fold increases).

6. From Figure 3c in this manuscript it seems that the drug has negative effects on NOR discrimination index in WT animals (together with the positive effects on CD mice). This should be discussed in the manuscript to emphasize the potential negative effects of the drug.

The effect referred by the Reviewer was a non-significant tendency towards a decrease of discrimination index in WT mice treated with JZL184. According to our results in the present manuscript, JZL184 shows surprisingly different effects in WT and CD mice, which could be due to a potential different basal status of the ECS. Therefore, we think that we cannot extrapolate the results from WT mice into CD mice, that are the target of the treatment. In this regard, we have emphasized the differential effect of the drug depending on the genetic context (CD or WT), since this is a common feature in several observations in the manuscript, as for example, we undoubtedly observed in the cardiac phenotype.

7. Why did the authors not characterize the signaling of the CB1R following drug treatment in Figure 3 as they did in Figure 2? This should be demonstrated, to further study the drug's effect on the ECS.

We thank the Reviewer for this important suggestion. We have now characterized CB1R G protein coupling after JZL184 treatment and we have demonstrated how CB1R-mediated Gi/o activity is normalized in specific brain regions of CD mice after treatment: basolateral amygdala, olfactory bulb and *CA1 stratum radiatum*. These results have been included in Figure 4, c and d.

8. The effect of the drug on the cardiovascular phenotype is not clear. There is no mechanistic explanation for this change. RNAseq is only partial aspect of this surprising effect, and is not sufficient to understand and support the mechanistic explanation of how the drug affected cardiovascular properties in only a few days. Why should it affect heart properties, especially given such a short treatment of only several days? This should be further demonstrated and explained, with more histological and molecular and cellular evidence to support this surprising effect. Moreover, did these changes in weight etc. were specific to the heart only? The different number of mice in each of the tests presented in Figure 4 is unjustified and makes the reviewer feel uncomfortable. Why did the author have to use 20 mice in Figure 4b, but used 7 in Figure 4c? Can they show statistical power calculation to support the huge difference in mice number between the tests?

Following the suggestions of the Reviewer on this point, we have obtained additional evidence to support the unexpected effect of JZ184 over the cardiovascular phenotype of CD mice.

First, we analyzed the size of cardiomyocytes in both CD and WT mice after JZL184 or vehicle treatment. The size of cardiomyocytes was found significantly increased in CD mice, and JZL184 treatment had a significant effect normalizing this anomalous parameter to WT values. This result further supports anti-hypertrophic effect of JZL184 over the cardiac phenotype of CD mice. These new data are included in Figure 5, b and c.

Second, we have now assessed systolic blood pressure of CD and WT mice exposed to JZL184 or vehicle treatment. CD mice showed an increase in arterial blood pressure that was reversed after JZL184 treatment (included in Figure 5e). Several studies have demonstrated that antihypertensive drugs cause a regression of left-ventricular hypertrophic phenotype (Van Zwieten, 2000), and reversion of similar cardiovascular phenotypes have been observed after different short treatments (15 days to 4 weeks) in other animal models (Marin et al., 2011; Molkentin et al., 1998; Ng et al., 2017).

Therefore, a decrease in blood pressure in CD mice may be playing a role in the normalization of the hypertrophic phenotype. This argument has been included in the discussion (Discussion, Lines 379-382) as a potential explanation of the rapid effect of JZL184 treatment over the hypertrophic phenotype.

Third, we assessed CB1R mRNA levels in the heart using qPCR approaches (Figure 5f). We observed a significant upregulation of CB1R mRNA in CD mice which was reversed after JZL184 treatment. These results suggest that enhanced CB1R density in the heart relate to the hypertrophic phenotype, and that JZL184 can abrogate both hypertrophy and CB1R overexpression in CD mice, without any effect in WT mouse heart tissues.

Fourth, we addressed whether the changes in weight by JZL184 were specific to the heart. CD mice show a significant decrease in body weight from the first month of age and it is maintained until two years of age (Segura-Puimedon et al., 2014). The weight of the brain is proportional to this decrease in body weight, but this is not the case for the cardiac tissue. Indeed, we measured brain weight normalized by body weight and we did not observe any changes due to JZL184 sub-chronic administration. These measures of relative brain weight are now displayed in Figure 5—figure supplement 2.

Regarding previous Figure 4b, commented by the Reviewer, it showed the mean of the 20 observations in 20 sections corresponding to 4-5 animals (4-5 sections per animal) as it was indicated in the figure legend. In the new version of the manuscript, we show these data averaging 4-5 observations for every mouse, and groups are composed by 4-5 mice (now Figure 5—figure supplement 1).

9. In Figure 4B, the muscle proportion measurement method in this paper is uncommon. Heart state whether in systolic or diastolic condition at the time of sacrifice and also paraffin processing possibly produce artifacts that will affect the heart morphology and ventricular wall thickness. It will be better if the researcher showed cardiac hypertrophy by echocardiography parameters such as PWTd (posterior wall thickness diastolic) PWTs (posterior wall thickness systolic).

As suggested by the Reviewer, we have now included additional echocardiographic data in Supplementary File 3 regarding heart morphology: left ventricular posterior wall thickness diastolic (LVPWd), interventricular septum diastolic (IVSd), left ventricular mass diastolic (Lvmass) and LV end-diastolic diameter (LVDd). In addition, as mentioned above, we have performed histological studies where we analyzed cardiomyocyte size in all experimental conditions and found significant phenotype effects that were resolved to control levels by treatment (Figure 5, b and c).

10. Since WBS cardiac phenotype includes hypertension, we suggest the researcher do blood pressure measurement in CD mice, also to confirm the phenotype from previous paper by Segura-Puimedon et al., Hum Mol Genet 2014.

We followed the suggestion of the Reviewer and assessed arterial blood pressure in CD and WT mice treated with vehicle or JZL184. Sub-chronic treatment of JZL184 reversed hypertension of CD mice. We have included these new results to Figure 5e.

11. Did the authors try acute or subchronic CB1 inverse agonists? If not, what would be the benefit of subchronic JZL184 than pharmacological CB1 inhibition? As a related discussion, do the authors expect any side (reverse) effect during the early phase of JZL184 treatment, maybe first a few days, when 2-AG signaling must have been elevated but changes in CB1 receptor expression is not occurred yet?

This is an interesting and valuable point. We did not assess the effect of acute or sub-chronic CB1R inverse agonists. This would be very interesting for future studies in CD mice. We included this possibility in the Discussion section (Discussion, Lines 397-401). We reasoned that the advantage of sub-chronic JZL184 treatment rather than pharmacological CB1R inhibition is that boosting the ECS with JZL184 may be particularly effective only on active sites of endocannabinoid production (Mechoulam and Parker, 2013), which could turn this approach more specific than a straight targeting of CB1R. Furthermore, pharmacological and genetic inactivation of MAGL has been demonstrated to exert anti-inflammatory properties under certain conditions by decreasing the synthesis of proinflammatory mediators, specifically, the eicosanoids derived from the arachidonic acid (Katz et al., 2014; Nomura et al., 2011; Pihlaja et al., 2015). Nomura et al., showed that in mice that have received lipopolysaccharide injection, JZL184 decreased inflammatory cytokine levels and hippocampal microglia did not show signs of activation. In addition, MAGL inactivation by JZL184 decreased hippocampal microglial reactivity in the hippocampus in a mouse model of Alzheimer’s disease (Pihlaja et al., 2015). Interestingly, it was recently described that CD mice show increased microglial reactivity in the hippocampus and motor cortex (Ortiz-Romero et al., 2021). Therefore, we cannot rule out, that the beneficial effects of JZL184 over CD mice may be in part mediated by these anti-inflammatory properties. Future studies should also address this question more specifically.

The possible effects during early phases of JZL184 treatment should be further explored in future studies. In the present study, we performed the Vsoc-maze for sociability and social novelty after a single administration of JZL184 (Figure 3—figure supplement 1) and we did not observe any significant effect of the treatment, while the repeated administration managed to improve the cognitive outcome. Then we focused the rest of the study on the effects of the sub-chronic schedule.

12. Endocannabinoid levels quantified from discrete brain regions closely related to the behavioral tests could be a good addition. Also, usually, the absolute values rather than comparative units are displayed for these kind of quantitation results (maybe shown in Table 1 is %? The unit was not clearly defined).

We have now analyzed the endocannabinoid levels of CD and WT mice in amygdala and hippocampus after administration of vehicle and of JZL184. We have not observed differences in endocannabinoids between CD and WT mice receiving vehicle. However, as expected, we observed the effect of JZL184 treatment over 2-AG levels in both WT and CD mice (Table 2). No changes were found in anandamide levels assessed in the same homogenates. Following the suggestion of the Reviewer, we have also changed relative values of Table 1 for absolute values.

13. Does JZL184-induced normalization CB1 receptors density observed in hippocampus or heart that are related to other phenotypes (NORT or cardiac hypertrophy, respectively) that were normalized by JZL184? Otherwise, is it possible that the beneficial effects are actually a result from long-term elevation of 2-AG signaling? If these results are not available, such limitation in interpretation should be discussed.

We now have assessed CB1R function after treatment in brain regions where CD mice presented CB1R activity alterations. Interestingly, we observed that CB1R activity is specifically normalized in the basolateral amygdala, the olfactory bulb and *CA1 stratum radiatum.* We cannot rule out the effect of long-term elevation of 2-AG signaling in those areas expressing monoacylglycerol lipase. Regarding CB1R in the heart, we now report mRNA levels of CB1R. We observed a decrease in CB1R mRNA expression of CD mice normalized after sub-chronic treatment of JZL184. Overall, CD mice present alterations in CB1R in different key regions and JZL184 sub-chronic treatment is normalizing some of these changes. These new data are displayed in Figure 5f.

References

Akkerman, S., Blokland, A., Reneerkens, O., van Goethem, N.P., Bollen, E., Gijselaers, H.J.M., Lieben, C.K.J., Steinbusch, H.W.M., and Prickaerts, J. (2012). Object recognition testing: Methodological considerations on exploration and discrimination measures. Behav. Brain Res. *232*, 335–347.

Berrendero, F., Romero, J., García-Gil, L., Suarez, I., De la Cruz, P., Ramos, J.A., and Fernández-Ruiz, J.J. (1998). Changes in cannabinoid receptor binding and mRNA levels in several brain regions of aged rats. Biochim. Biophys. Acta *1407*, 205–214.

Chanda, P.K., Gao, Y., Mark, L., Btesh, J., Strassle, B.W., Lu, P., Piesla, M.J., Zhang, M.Y., Bingham, B., Uveges, A., et al. (2010). Monoacylglycerol lipase activity is a critical modulator of the tone and integrity of the endocannabinoid system. Mol. Pharmacol. *78*, 996–1003.

Collins, R.T. (2018). Cardiovascular disease in Williams syndrome. Curr. Opin. Pediatr. *30*, 609–615.

Ginsburg, B.C., and Hensler, J.G. (2022). Age-related changes in CB1 receptor expression and function and the behavioral effects of cannabinoid receptor ligands. Pharmacol. Biochem. Behav. *213*, 173339.

Gomis-González, M., Galera-López, L., Ten-Blanco, M., Busquets-Garcia, A., Cox, T., Maldonado, R., and Ozaita, A. (2021). Protein Kinase C-Γ Knockout Mice Show Impaired Hippocampal Short-Term Memory While Preserved Long-Term Memory. Mol. Neurobiol. *58*, 617–630.

Katz, P.S., Sulzer, J.K., Impastato, R.A., Teng, S.X., Rogers, E.K., and Molina, P.E. (2014). Endocannabinoid Degradation Inhibition Improves Neurobehavioral Function, Blood–Brain Barrier Integrity, and Neuroinflammation following Mild Traumatic Brain Injury. Https://Home.Liebertpub.Com/Neu *32*, 297–306.

Kozel, B.A., Barak, B., Kim, C.A., Mervis, C.B., Osborne, L.R., Porter, M., and Pober, B.R. (2021). Williams syndrome. Nat. Rev. Dis. Prim. *7*.

Liu, P., Bilkey, D.K., Darlington, C.L., and Smith, P.F. (2003). Cannabinoid CB1 receptor protein expression in the rat hippocampus and entorhinal, perirhinal, postrhinal and temporal cortices: regional variations and age-related changes. Brain Res. *979*, 235–239.

Maccarrone, M., Valverde, O., Barbaccia, M.L., Castañé, A., Maldonado, R., Ledent, C., Parmentier, M., and Finazzi-Agrò, A. (2002). Age-related changes of anandamide metabolism in CB1 cannabinoid receptor knockout mice: Correlation with behaviour. Eur. J. Neurosci. *15*, 1178–1186.

Marin, T.M., Keith, K., Davies, B., Conner, D.A., Guha, P., Kalaitzidis, D., Wu, X., Lauriol, J., Wang, B., Bauer, M., et al. (2011). Rapamycin reverses hypertrophic cardiomyopathy in a mouse model of LEOPARD syndrome–associated PTPN11 mutation. J. Clin. Invest. *121*, 1026.

Mechoulam, R., and Parker, L.A. (2013). No Title. *64*, 21–47.

Molkentin, J.D., Lu, J.R., Antos, C.L., Markham, B., Richardson, J., Robbins, J., Grant, S.R., and Olson, E.N. (1998). A Calcineurin-Dependent Transcriptional Pathway for Cardiac Hypertrophy. Cell *93*, 215–228.

Navarro-Romero, A., Vázquez-Oliver, A., Gomis-González, M., Garzón-Montesinos, C., Falcón-Moya, R., Pastor, A., Martín-García, E., Pizarro, N., Busquets-Garcia, A., Revest, J.-M., et al. (2019). Cannabinoid type-1 receptor blockade restores neurological phenotypes in two models for Down syndrome.

Ng, H.H., Leo, C.H., Prakoso, D., Qin, C., Ritchie, R.H., and Parry, L.J. (2017). Serelaxin treatment reverses vascular dysfunction and left ventricular hypertrophy in a mouse model of Type 1 diabetes. Sci. Rep. *7*.

Nomura, D.K., Morrison, B.E., Blankman, J.L., Long, J.Z., Kinsey, S.G., Marcondes, M.C.G., Ward, A.M., Hahn, Y.K., Lichtman, A.H., Conti, B., et al. (2011). Endocannabinoid hydrolysis generates brain prostaglandins that promote neuroinflammation. Science *334*, 809–813.

Ortiz-Romero, P., González-Simón, A., Egea, G., Pérez-Jurado, L.A., and Campuzano, V. (2021). Co-Treatment With Verapamil and Curcumin Attenuates the Behavioral Alterations Observed in Williams-Beuren Syndrome Mice by Regulation of MAPK Pathway and Microglia Overexpression. Front. Pharmacol. *12*.

Pihlaja, R., Takkinen, J., Eskola, O., Vasara, J., López-Picón, F.R., Haaparanta-Solin, M., and Rinne, J.O. (2015). Monoacylglycerol lipase inhibitor JZL184 reduces neuroinflammatory response in APdE9 mice and in adult mouse glial cells. J. Neuroinflammation *12*, 1–6.

Royston, R., Waite, J., and Howlin, P. (2019). Williams syndrome: recent advances in our understanding of cognitive, social and psychological functioning. Curr. Opin. Psychiatry *32*, 60–66.

Schlosburg, J.E., Blankman, J.L., Long, J.Z., Nomura, D.K., Pan, B., Kinsey, S.G., Nguyen, P.T., Ramesh, D., Booker, L., Burston, J.J., et al. (2010). Chronic monoacylglycerol lipase blockade causes functional antagonism of the endocannabinoid system. Nat. Neurosci. *13*, 1113–1119.

Segura-Puimedon, M., Sahún, I., Velot, E., Dubus, P., Borralleras, C., Rodrigues, A.J., Valero, M.C., Valverde, O., Sousa, N., Herault, Y., et al. (2014). Heterozygous deletion of the Williams-Beuren syndrome critical interval in mice recapitulates most features of the human disorder. Hum. Mol. Genet. *23*, 6481–6494.

Wang, L., Liu, J., Harvey-White, J., Zimmer, A., and Kunos, G. (2003). Endocannabinoid signaling via cannabinoid receptor 1 is involved in ethanol preference and its age-dependent decline in mice. Proc. Natl. Acad. Sci. U. S. A. *100*, 1393–1398.

Van Zwieten, P.A. (2000). The influence of antihypertensive drug treatment on the prevention and regression of left ventricular hypertrophy. Cardiovasc. Res. *45*, 82–91.